# Cysteine synthases CYSL-1 and CYSL-2 mediate *C. elegans* heritable adaptation to *P. vranovensis* infection

Nicholas O. Burton [1✉], Cristian Riccio[2,3], Alexandra Dallaire [2,4], Jonathan Price[2,4], Benjamin Jenkins[5], Albert Koulman [5] & Eric A. Miska [2,3,4]

Parental exposure to pathogens can prime offspring immunity in diverse organisms. The mechanisms by which this heritable priming occurs are largely unknown. Here we report that the soil bacteria *Pseudomonas vranovensis* is a natural pathogen of the nematode *Caenorhabditis elegans* and that parental exposure of animals to *P. vranovensis* promotes offspring resistance to infection. Furthermore, we demonstrate a multigenerational enhancement of progeny survival when three consecutive generations of animals are exposed to *P. vranovensis*. By investigating the mechanisms by which animals heritably adapt to *P. vranovensis* infection, we found that parental infection by *P. vranovensis* results in increased expression of the cysteine synthases *cysl-1* and *cysl-2* and the regulator of hypoxia inducible factor *rhy-1* in progeny, and that these three genes are required for adaptation to *P. vranovensis*. These observations establish a CYSL-1, CYSL-2, and RHY-1 dependent mechanism by which animals heritably adapt to infection.

[1] Centre for Trophoblast Research, Department of Physiology, Development and Neuroscience, University of Cambridge, Cambridge CB2 3EG, UK. [2] Gurdon Institute, University of Cambridge, Cambridge CB2 1QN, UK. [3] Wellcome Sanger Institute, Wellcome Genome Campus, Cambridge CB10 1SA, UK. [4] Department of Genetics, University of Cambridge, Downing Street, Cambridge CB2 3EH, UK. [5] Metabolic Research Laboratories, University of Cambridge, Cambridge CB2 0QQ, UK. ✉email: nob20@cam.ac.uk

Multigenerational responses to environmental stress have been reported in evolutionarily diverse organisms[1–7]. These studies investigated a variety of types of stresses ranging from osmotic stress[1], to mitochondrial stress[5], to pathogen infection[8]. In each case, parental exposure to stress appeared to prime offspring to respond to a similar stress. For example, studies of *Arabadopsis thaliana* have demonstrated that parental exposure to mild osmotic stress can promote offspring resistance to future osmotic stress[9]. While many multigenerational effects of the environment have been described in plants and invertebrates, similar observations have recently been extended to vertebrates, including mammals. For example, parental exposure to high population density was demonstrated to promote an accelerated postnatal growth rate in red squirrels that enhanced offspring survival by allowing them to acquire territories more quickly[2]. Collectively, these findings raise the exciting possibility that parental exposure to environmental stress causing programmed changes in offspring physiology might represent a fundamental and significantly understudied aspect of inheritance with implications for diverse fields of biological and medical sciences.

Among the diverse environmental stresses an organism might encounter, pathogens, such as viruses, bacteria, and some eukaryotes are among the most ubiquitous stresses found in nature[10]. To adapt to the constant threat of pathogens, many organisms have evolved mechanisms by which parental exposure to pathogens can prime offspring immunity[6,11–16]. For example, in mammals, a mother can transfer specific antibodies to her offspring via milk to prime offspring immunity[16]. Similar observations of parents priming offspring immunity in response to pathogens have been reported in both plants[15] and invertebrates[11], even though these organisms lack antibodies. These findings suggest that multiple independent mechanisms have evolved for parents to prime offspring immunity.

The mechanisms by which parental exposure to pathogens might result in adaptive changes in offspring in organisms that lack antibodies remain largely unknown. Here we identify that *Caenorhabditis elegans* can heritably adapt to a natural pathogen, *Pseudomonas vranovensis*, and that adaptation to *P. vranovensis* requires the cysteine synthases CYSL-1 and CYSL-2 and the regulator of hypoxia inducible factor RHY-1.

## Results

**Identification of *P. vranovensis* as a pathogen of *C. elegans*.** One of the major obstacles in determining the molecular mechanisms underlying the multigenerational effects of pathogen infection is the lack of robust models that can be easily analyzed in the laboratory. To address this obstacle, we sought to develop a robust model of the heritable effects of bacterial infection in the nematode *C. elegans* by testing whether parental infection of *C. elegans* with bacterial pathogens from its natural environment could affect offspring response to future infection. Previous sampling of bacterial species from the natural environments of *C. elegans* identified 49 as yet undescribed bacterial isolates that induce the expression of the immune response genes *irg-1* or *irg-5*[17]. We found that 24 h of parental exposure to two of the isolates, BIGb446 and BIGb468, significantly enhanced offspring survival in response to future exposure to the same bacteria (Fig. 1a, b). Specifically, ~95% of newly hatched larvae from parents fed a standard laboratory diet of *E. coli* HB101 died within 24 h of hatching on NGM agar plates seeded with BIGb446 or BIGb468 (Fig. 1a, b). By contrast, 45% of embryos from adults exposed to BIGb446 or BIGb468 were still alive at 24 h (Fig. 1a, b) and a majority of these larvae survived through adulthood (Fig. 1c, d). We conclude that parental exposure to bacterial isolates BIGb446

and BIGb468 enhances offspring survival in response to future exposure to these bacteria.

To determine the species identity of bacterial isolates BIGb446 and BIGb468 we performed long read whole genome sequencing and assembled the genomes of these bacteria. We found that the genomes of both BIGb446 and BIGb468 were ~5.9 Mb (Supplementary Data 1 and 2) and were 99.38% identical across the entire genome (Supplementary Data 3 and Supplementary Fig. 1). We concluded that BIGb446 and BIGb468 are isolates of the same species. We compared the 16S rRNA sequence of BIGb446 to known bacterial genomes using BLAST. We found that the 16S rRNA sequence from BIGb446 is 99.93% identical to *P. vranovensis* strain 15D11. Previous studies of *Pseudomonas* phylogeny have used the DNA sequences of *gyrB*, *rpoB*, and *recA* to differentiate species of *Pseudomonas*, with similarity above 97% set as the accepted species threshold[18,19]. We found that the sequence of *gyrB* was 98.05% identical, the sequence of *rpoB* was 99.44% identical, and the sequence of *recA* was 98.66% identical to the sequences of homologous genes in *P. vranovensis* strain 15D11. No other species of *Pseudomonas* was >97% identical to *Pseudomonas* sp. BIGb446 at any of these three genes. Furthermore, we compared the average nucleotide identity (ANI) of our assembly of *Pseudomonas* sp. BIGb446 with the genome of *P. vranovensis* strain 15D11 using OrthoANIu[20]. We found these two genomes had an ANI of 97.33%. We conclude that *Pseudomonas* sp. BIGb446 and *Pseudomonas* sp. BIGb468 are isolates of *P. vranovensis*, a Gram-negative soil bacteria[21].

To test if live bacteria were required for killing, we exposed animals to UV-killed *P. vranovensis*. We found that UV-killed *P. vranovensis* did not cause any observable lethality (Fig. 1e), indicating that live bacteria are required for killing. Similarly, to test if live bacteria were required for the observed adaptive effect of parental exposure to *P. vranovensis* on offspring survival, we transferred *P. vranovensis* exposed adults to plates containing antibiotics for 24 h. We found that parental antibiotic treatment eliminated the enhancement of offspring survival in response to future *P. vranovensis* exposure (Supplementary Fig. 2a). We conclude that parental exposure to live *P. vranovensis* is required to enhance offspring survival in response to future *P. vranovensis* exposure and that eliminating the infection in adults also eliminates the adaptation in offspring.

Collecting embryos by bleaching results in a pool of mixed stage embryos, ranging from one-cell embryos to three-fold stage embryos. We hypothesized that differences in embryo staging might underlie the observed adaptation. To test this, we individually picked 100 three-fold stage embryos laid by adults fed either *E. coli* or *P. vranovensis*. We found that 3 out of 100 three-fold stage embryos from parents fed *E. coli* were alive 24 h after hatching on plates seeded with *P. vranovensis* (Supplementary Fig. 2b). By contrast, we found that 32 out of 100 three-fold stage embryos from adults exposed to *P. vranovensis* were alive 24 h after hatching on plates seeded with *P. vranovensis* (Supplementary Fig. 2b). These results are similar to our observations when embryos are collected by bleaching. We conclude that differences in embryo staging do not underlie the observed heritable adaptation.

Our data indicate that parental (P0) exposure to *P. vranovensis* can enhance offspring (F1) survival as newly hatched larvae (Fig. 1). To test if parental exposure to *P. vranovensis* can also enhance F1 adult survival in response to *P. vranovensis* we compared the survival of F1 adults from parents fed *E. coli* to the survival of F1 adults from parents exposed to *P. vranovensis*. We found that naïve adults are significantly more resistant to *P. vranovensis* than L1-stage larva, surviving up to 72 h of exposure to the pathogen, and that parental exposure to *P. vranovensis* did

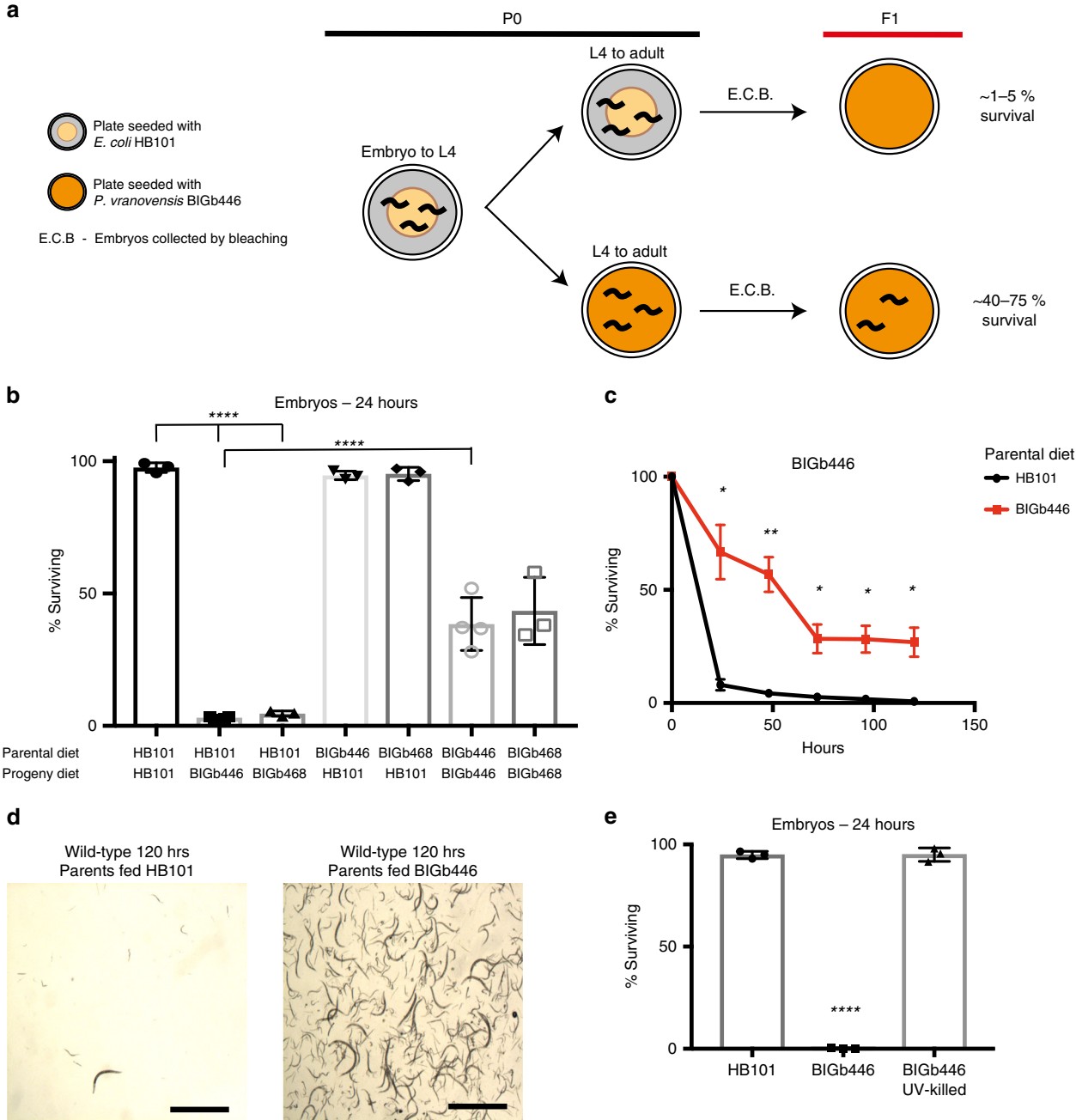

**Fig. 1 *C. elegans* heritably adapts to infection by *Pseudomonas vranovensis*. a** Graphical representation of experimental set up. Embryos collected by bleaching were placed onto fresh plates seeded with BIGb446 immediately after egg prep and the percent of surviving animals was counted after 24 h. **b** Percent of wild-type animals surviving on plates seeded with either *E. coli* HB101 or bacterial isolates BIGb446 and BIGb468 after 24 h. Data presented as mean values ± s.d. *n* = 3–4 experiments of >100 animals. **c** Percent of wild-type animals surviving on plates seeded with bacterial isolate BIGb446. Data presented as mean values surviving at each time point from three separate experiments ± s.d. *n* = 3 experiments of 500 animals. Log-rank test of individual Kaplan–Meier survival curves *p* < 0.001. **d** Images of wild-type animals surviving after 120 h of feeding on bacterial isolate BIGb446. 1000 animals were used at *t* = 0 in each condition and surviving animals were resuspended in 20 μl M9 and imaged. Scale bars 1 mm. Experiment repeated three times with similar results. **e** Percent of wild-type animals surviving on *E. coli* HB101 or bacterial isolate BIGb446 after 24 h. Data presented as mean values ± s.d. *n* = 3 experiments of >100 animals. \**p* < 0.05, \*\**p* < 0.01, \*\*\*\**p* < 0.0001. Source data are provided as a Source Data file. See statistics and reproducibility section for statistical tests run.

not enhance F1 adult survival in response to *P. vranovensis* (Supplementary Fig. 2c). We conclude that parental exposure to *P. vranovensis* does not enhance F1 adult survival in response to *P. vranovensis* when F1 larvae develop on the non-pathogenic food source *E. coli* HB101. This observation is similar to our finding that clearing the infection in adults using antibiotics also eliminated the observed adaptation in offspring. We hypothesize

that clearing the infection in parents also results in losing the observed adaptive effect in offspring.

In some cases, the effects of parental stress on offspring have been reported to be intergenerational and only last a single generation[1]. In other cases, the effects of parental stress on offspring have been reported to persist transgenerationally and thus affect descendants several generations later[5,8,22]. To test

whether exposure to *P. vranovensis* has any transgenerational effects on immunity we first fed adult animals *P. vranovensis* for 24 h and assayed the response of progeny one and two generations later. We found that a single exposure of adult animals to *P. vranovensis* enhanced their F1 offspring's survival (Fig. 2a) but did not enhance the survival of their F2 descendants two generations later (Fig. 2a). These results indicate that a single generation of exposure to *P. vranovensis* only intergenerationally affects progeny survival.

**Identification of a multigenerational response to *P. vranovensis*.** Some studies of the effects of parental environment on offspring have found that multiple consecutive generations of exposure of animals to the same stress can enhance adaptation to stress in descendants[5]. We therefore tested whether the exposure of five consecutive generations (P0, F1, F2, F3, and F4) of animals to *P. vranovensis* could enhance F5 progeny survival in response to

*P. vranovensis* infection but found that F5 survival in this case was not significantly different from F1 survival after a single generation of exposure to *P. vranovensis* (Fig. 2a). These results are consistent with a model where *P. vranovensis* infection intergenerationally affects offspring survival and are similar to other intergenerational adaptations in *C. elegans*, such as the observation that maternal exposure to osmotic stress protects larvae via a mechanism dependent on insulin-like signaling to oocytes[1]. Taken together, these observations suggest that intergenerational programming of offspring adaptations to environmental stresses, such as osmotic stress and *P. vranovensis* infection, might be more common than currently appreciated.

Our observations suggest that in most cases *C. elegans* exposure to *P. vranovensis* only intergenerationally affects offspring survival, possibly via maternal effects, and resets to a "naïve" response if animals are removed from the pathogen. However, we noted that in some observations of transgenerational effects, including transgenerational responses to pathogen infection[23],

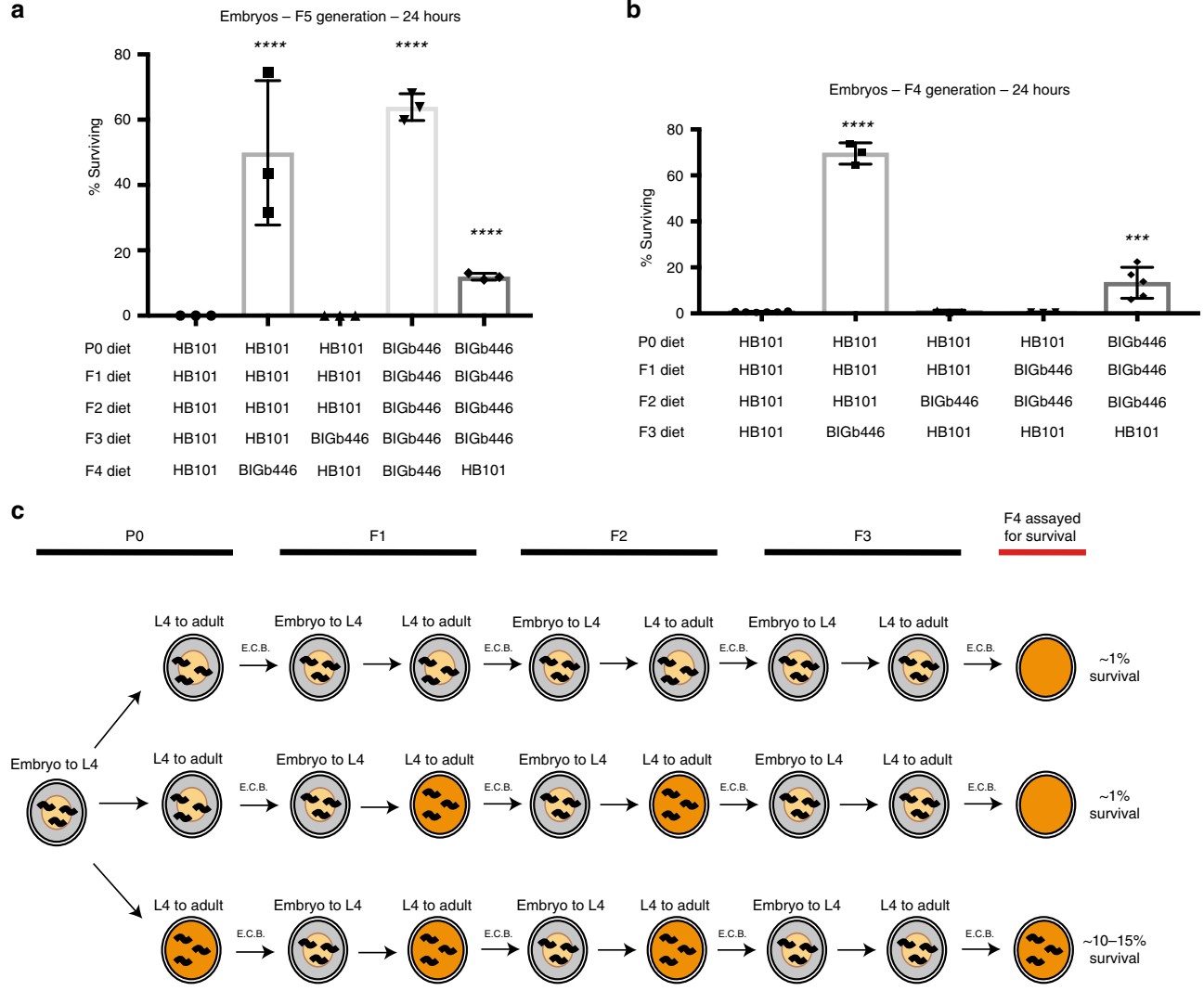

**Fig. 2 *C. elegans* exposure to *P. vranovensis* can have multigenerational effects. a** Percent of wild-type animals surviving on plates seeded with bacterial isolate BIGb446 after 24 h. Data presented as mean values ± s.d. *n* = 3 experiments of >100 animals. **b** Percent of wild-type animals surviving on plates seeded with bacterial isolate BIGb446 after 24 h. Data presented as mean values ± s.d. *n* = 3–6 experiments of 500 animals each. Wild-type animals carried the integrated transgene *nls470*. **c** Graphical representation of multigenerational experimental set up. See Fig. 1a for legend. Embryos were collected by bleaching (E.C.B.) in each generation and were placed onto fresh plates seeded with *E. coli* HB101 until the L4 developmental stage at each generation until the final generation, where animals were placed onto fresh plates seeded with *P. vranovensis* BIGb446 and the percent of surviving animals was counted after 24 h. Source data are provided as a Source Data file. ****p* = 0.003, *****p* < 0.0001. See section "statistics and reproducibility" for statistical tests run.

the transgenerational effects only appeared after multiple consecutive generations of exposure of animals to pathogen infection[23]. We therefore also tested whether the exposure of multiple consecutive generations of animals to *P. vranovensis* could cause animals' resistance to *P. vranovensis* to persist for more than a single generation. We found that, unlike a single generation of exposure to *P. vranovensis*, the exposure of four consecutive generations of animals (P0, F1, F2, and F3) to *P. vranovensis* enhanced the survival of F5 animals, even when their F4 parents were never exposed to *P. vranovensis* (Fig. 2a). These data indicate that multiple consecutive generations of exposure to *P. vranovensis* has different effects on the survival of descendants when compared to a single generation of exposure.

To determine how many consecutive generations of exposure to *P. vranovensis* were required for the adaptive effect to persist for more than one generation, we exposed three (P0, F1, and F2), two (F1 and F2), and one (F2) generation of animals to *P. vranovensis* and assayed the survival of F4 generation animals exposed to *P. vranovensis* (Fig. 2b, c). We found that one generation of exposure (F2 exposure) and two consecutive generations of exposure to *P. vranovensis* (F1 and F2 exposure) did not affect F4 survival (Fig. 2b). By contrast, we found that three consecutive generations of exposure to *P. vranovensis* (P0, F1, and F2 exposure) significantly enhanced the survival of F4 generation animals (Fig. 2b, c). These results indicate that the exposure of P0 animals to *P. vranovensis* is required to enhance the survival of F4 generation animals under conditions where F1 and F2 animals are also exposed to *P. vraonvensis* (Fig. 2c). Our findings suggest that similar multigenerational effects, while often not investigated, might be more common than currently appreciated.

**Comparison to previous observations of heritable effects**. To better understand the mechanisms by which *C. elegans* can heritably adapt to *P. vranovensis* we investigated how parental exposure to *P. vranovensis* can lead to a between 10 and 50-fold increase in offspring survival in response to future *P. vranovensis* exposure. As a first approach, we tested whether mutations in any factors previously reported to be involved in intergenerational or transgenerational responses to stress in *C. elegans* were required for our observed heritable adaptation to *P. vranovensis*, including small RNA-mediated pathways (*hrde-1*[24], *prg-1*[8]), H3K9 methylation (*set-32*[25] and *met-2*[4]), H3K4 methylations (*set-2, wdr-5.1, spr-5*)[5,22], DNA adenosine methylation (*damt-1*)[5], and the RAS/ERK-signaling pathway (*lin-45*)[1]. We found that none of these factors were required for *C. elegans* adaptation to *P. vranovensis* (Supplementary Fig. 3a). These data suggest that *C. elegans'* adaptation to *P. vranovensis* involves an as-yet-unknown mechanism.

Bacterial isolates BIGb446 and BIGb468 were originally reported to promote the expression of immune response gene *irg-5*[17]. The expression of *irg-5* in response to pathogen infection is controlled by the p38-like MAP kinase PMK-1[26]. We tested whether PMK-1 is required for resistance to *P. vranovensis* by placing wild-type and *pmk-1* mutant adults on plates seeded with *P. vranovensis*. We found that 100% of *pmk-1* mutants were dead after 48 h while more than 40% of wild-type animals remained alive (Supplementary Fig. 3b). These results indicate that PMK-1 promotes resistance to *P. vranovensis*. We also tested whether PMK-1 was required for animals to heritably adapt to *P. vranovensis* by exposing wild-type and *pmk-1* mutants to *P. vranovensis* for 24 h and assaying the survival of their offspring in response to repeated exposure to *P. vranovensis*. We found that PMK-1 is not required for the heritable adaptation to *P. vranovensis* (Supplementary Fig. 3c).

These data suggest that signaling via PMK-1 has some effect on *C. elegans* response to *P. vranovensis* but is likely not mediating *C. elegans'* adaptation to *P. vranovensis*.

Previous studies found that *C. elegans* can heritably adapt to osmotic stress by regulating metabolism and gene expression in offspring[1,27]. To test whether parental exposure to *P. vranovensis* results in similar metabolic changes in offspring as parental exposure to osmotic stress we profiled 92 lipid metabolites that were previously observed to heritably change in abundance in response to osmotic stress by LC/MS. We found that only six metabolites exhibited modest changes in abundance in embryos from parents exposed to *P. vranovensis* when compared to embryos from parents fed *E. coli* HB101 (Supplementary Fig. 3d and Supplementary Data 4). We conclude that parental exposure to *P. vranovensis* does not result in similar changes in offspring metabolism as parental exposure to osmotic stress[1], and that these heritable adaptations to environmental stress are likely to be distinct.

To determine if parental infection by *P. vranovensis* affects offspring gene expression we profiled mRNA abundance by RNA-seq. We first quantified gene expression in P0 adults exposed to *P. vranovensis*. We found that 1275 genes exhibited a greater than two-fold change in expression when compared to adults fed *E. coli* HB101 (Supplementary Data 5). We then profiled gene expression in F1 embryos from parents exposed to *P. vranovensis*. We found that 1644 genes exhibited a greater than two-fold change in expression in embryos from parents exposed to *P. vranovensis* when compared to embryos from parents fed *E. coli* HB101 (Fig. 3a and Supplementary Data 5). Of these 1644 genes that changes in expression in F1 embryos, 398 exhibited a similar change in P0 adults exposed to *P. vranovensis* (Fig. 3b). We conclude that animals exhibit significant changes in gene expression in response to *P. vranovensis* and that ~30% of the changes in gene expression observed in infected parents are also observed in F1 embryos.

To further investigate the gene expression changes that occur in response to *P. vranovensis*, we compared a list of the 100 genes exhibiting the largest increase in expression in F1 embryos from parents exposed to *P. vranovensis* to all published gene expression profiles in *C. elegans* using the web-based application WormExp v1.0[28]. When comparing genes regulated by *P. vranovensis* to genes regulated by other microbes, we found that *C. elegans* response to *P. vranovensis* is most similar to *C. elegans* response to the pathogen *Photorhabdus luminescens*[29], with 46 out of 100 genes upregulated in response to both pathogens, and to the pathogen *Pseudomonas aeruginosa*[30], with 28 of 100 genes upregulated by both pathogens[31]. These observations are consistent with *P. vranovensis* being a pathogen of *C. elegans*.

*C. elegans'* heritable adaptation to *P. vranovensis* might be a general response to bacterial pathogens or specific to *P. vranovensis*. To test whether parental exposure to pathogens that activate a similar transcriptional response can also protect offspring from *P. vranovensis*, we exposed young adult animals to *P. aeruginosa* PA14 and *P. luminescens* Hb and assayed the response of their offspring to *P. vranovensis*. We found that parental infection by these two pathogens did not enhance offspring survival in response to *P. vranovensis* (Fig. 3c). These results indicate that adaptation to *P. vranovensis* is not a generic response to pathogenic bacteria.

We hypothesized that parental exposure to different bacterial infections might not protect offspring from *P. vranovensis* infection because they do not cause all of the specific changes in gene expression in offspring that are required for adaptation. To compare how parental infection by different bacterial pathogens affects offspring gene expression we exposed young adults to *P. aeruginosa* or *P. luminescens* for 24 h and collected

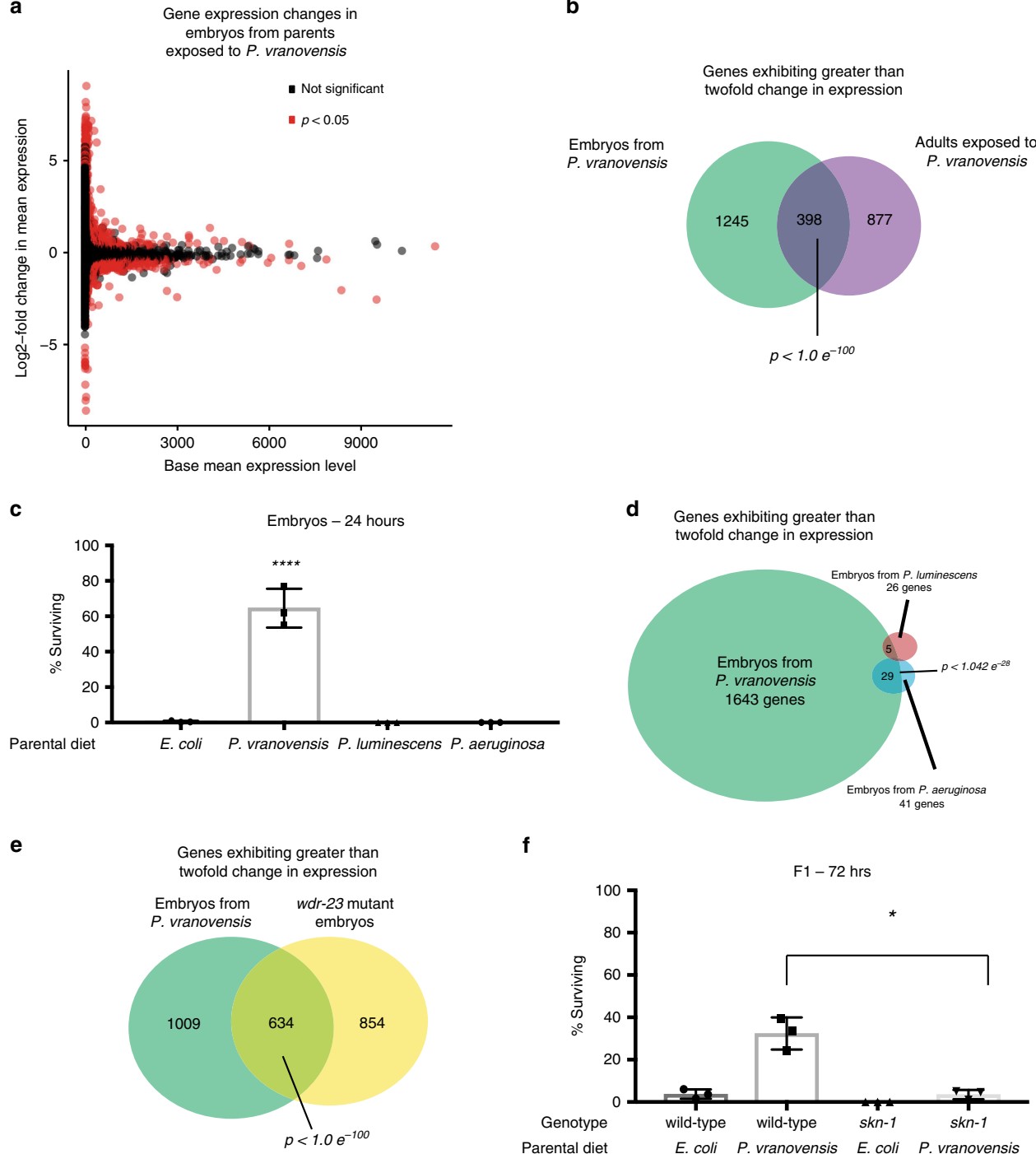

**Fig. 3 Parental infection by *P. vranovensis* alters gene expression in offspring. a** Gene expression changes in embryos from parents fed *P. vranovensis* BIGb446 when compared to embryos from parents fed *E. coli* HB101. Values represent averages from three replicates. **b** Venn diagram of genes exhibiting a greater than two-fold change in RNA expression in adults fed *P. vranovensis* BIGb446 and embryos from adults fed *P. vranovensis* BIGb446. *p*-value represents normal approximation to the hypergeometric probability (see section "Statistics and reproducibility"). **c** Percent of F1 wild-type animals surviving on plates seeded with bacterial isolate BIGb446 after 24 h. P0 animals were fed their normal laboratory diet of *E. coli* HB101 or were exposed to the bacterial pathogens *P. vranovensis* BIGb446, *P. aeruginosa* PA14, or *P. luminescens* Hb for 24 h. Data presented as mean values ± s.d. *n* = 3 experiments of 500 animals. **d** Venn diagram of genes exhibiting a greater than two-fold change in RNA expression in embryos from parents fed *P. vranovensis* BIGb446, *P. aeruginosa* PA14, and *P. luminescens* Hb. *p*-value represents normal approximation to the hypergeometric probability (see section "Statistics and reproducibility"). **e** Venn diagram of number of genes exhibiting a greater than two-fold change in RNA expression in embryos from parents exposed to *P. vranovensis* BIGb446 and *wdr-23(mac32)* mutant embryos. *p*-value represents normal approximation to the hypergeometric probability (see section "Statistics and reproducibility"). **f** Percent of wild-type and *skn-1(zj15)* mutants surviving on plates seeded with *P. vranovensis* BIGb446 after 72 h. Error bars, s.d. *n* = 3 experiments of >100 animals. Source data are provided as a Source Data file. *\*p* = 0.0057, *\*\*\*\*p* < 0.0001. See section "Statistics and reproducibility" for statistical tests run.

embryos from these animals to profile gene expression by RNA-seq. We found that only 41 genes exhibited a greater than two-fold change in expression in embryos from parents exposed to *P. aeruginosa* when compared to embryos from parents fed *E. coli* HB101 and of these genes only 29 also exhibited altered gene expression in embryos from parents exposed to *P. vranovensis* (Fig. 3d and Supplementary Data 6). Separately, we found that only 26 genes exhibited a greater than two-fold change in expression in embryos from parents exposed to *P. luminescens* when compared to parents fed *E. coli* HB101 (Fig. 3d and Supplementary Data 6). Of these genes, only five also exhibited altered expression in embryos from parents exposed to *P. vranovensis* (Fig. 3d). Collectively, these results indicate that parental infection by *P. aeruginosa* and *P. luminescens* have only a limited effect on offspring gene expression when compared to parental infection by *P. vranovensis*. We conclude that parental exposure to different pathogenic diets has distinct effects on offspring gene expression and speculate that these differences might explain why parental exposure to *P. vranovensis* results in offspring resistance to future exposure to *P. vranovensis* while parental exposure to other pathogens does not.

In an attempt to identify potential signaling pathways that might regulate *C. elegans* adaptation to *P. vranovensis* we also compared a list of the 100 genes exhibiting the largest changes in expression in F1 embryos from parents exposed to *P. vranovensis* to expression data from all previously characterized mutant animals using WormExp v1.0[28]. We found that *C. elegans* response to *P. vranovensis* is most similar to *wdr-23* mutant animals, with nearly half (51 of 105) of genes showing abnormal expression in both conditions[32]. To confirm these findings, we performed RNA-seq of *wdr-23* mutant embryos. We found that 1488 genes exhibit a greater than two-fold change in expression in *wdr-23* mutant embryos, of which 634 also exhibited a greater than two-fold change in expression in response to *P. vranovensis* (Fig. 3e and Supplementary Data 7). These results indicate that the transcriptional response to *P. vranovensis* is very similar to the transcriptional changes found in *wdr-23* mutants.

**SKN-1 is required for adaptation to *P. vranovensis*.** WDR-23 regulates the expression of stress response genes by inhibiting the conserved stress response transcription factor SKN-1 and many of the gene expression changes in *wdr-23* mutants are thought to be the result of a constitutively activated SKN-1[32]. Furthermore, SKN-1 was previously demonstrated to regulate a subset of *C. elegans'* response to multiple different pathogens, including *P. aeruginosa* and *Enterococcus faecalis*[33]. These results suggested that signaling via SKN-1, rather than PMK-1, might regulate *C. elegans* adaptation to *P. vranovensis*. To test this hypothesis, we exposed partial loss-of-function *skn-1* mutant adults (null mutants are embryonic lethal) to *P. vranovensis* and assayed the survival of their offspring in response to *P. vranovensis*. We found that partial loss-of-function *skn-1* mutants exhibited modestly reduced adaptation to *P. vranovensis* when F1 offspring survival was assayed at 24 h (Supplementary Fig. 3e) and significantly reduced adaptation when F1 offspring survival was assayed after 72 h of exposure to *P. vranovensis* (Fig. 3f). We conclude that proper signaling via SKN-1 is required for *C. elegans* adaptation to *P. vranovensis*.

Differences in gene expression in F1 embryos from parents fed *E. coli* or exposed to *P. vranovensis* might be due to either differences in developmental staging of embryos from stressed parents or due to differences in SKN-1 target gene expression caused by *P. vranovensis* exposure. To differentiate between these two possibilities, we profiled the expression of the glutathione-S-transferase *gst-31*, which is one of the most upregulated genes in

both F1 embryos from parents exposed to *P. vranovensis* and in *wdr-23* mutant embryos, specifically in similarly staged (three-fold) embryos using a GFP reporter. We found that *gst-31* exhibits little to no expression in embryos from parents fed *E. coli* but is robustly expressed throughout three-fold stage embryos from parents exposed to *P. vranovensis* (Supplementary Fig. 4a, b). These data indicate that parental exposure to *P. vranovensis* can cause changes in F1 embryo gene expression that are not due to differences in developmental staging and suggest that SKN-1 target gene expression in embryos might be regulated by parental pathogen exposure.

**CYSL-1, CYSL-2, and RHY-1 are required for adaptation.** Among the genes exhibiting increased expression in response to *P. vranovensis* we identified the cysteine synthases *cysl-1* and *cysl-2* (Fig. 4a, b and Supplementary Data 5). CYSL-1 and CYSL-2 were previously reported to be involved in breaking down bacterial toxins produced by *P. aeruginosa*[34], and we found that the expression of both genes is upregulated in *wdr-23* mutant embryos (Supplementary Data 7). Based on these findings, we hypothesized that *cysl-1* and *cysl-2* might be required for adaptation to *P. vranovensis*. We confirmed that exposure of parents to *P. vranovensis* increased *cysl-2* expression in embryos using a GFP reporter (Fig. 4c)[35]. We then assayed seven independent mutants lacking *cysl-1* and two independent mutants lacking *cysl-2* and found that these mutants were unable to adapt to *P. vranovensis* (Fig. 4d, e and Supplementary Fig. 4c). In addition, we found that the defect caused by loss of *cysl-2* could be rescued by expressing a wild-type copy of *cysl-2* (Fig. 4e). We conclude that CYSL-1 and CYSL-2 are required for adaptation to *P. vranovensis*.

To further investigate the expression of *cysl-1* and *cysl-2* in response to *P. vranovensis* we examined transgenic animals expressing GFP under the control of the *cysl-1* and *cysl-2* promoters. We found that adult animals fed a normal laboratory diet of *E. coli* HB101 express *cysl-1* predominately in neurons with faint expression in hypodermal cells (Supplementary Fig. 5a) and *cysl-2* predominately in the pharynx with faint expression in hypodermal cells (Supplementary Fig. 5b). We did not observe significant changes in *cysl-1::GFP* expression in response to *P. vranovensis* in either P0 adults or F1 embryos, possibly because multicopy transgenic animals already highly overexpress *cysl-1:: GFP* (Supplementary Fig. 5a, c). However, we did observe strong induction of *cysl-2::GFP* in many tissues in both P0 adults and F1 embryos in response to *P. vranovensis*, with expression most strongly observed in hypodermal cells (Supplementary Fig. 5b, d). *cysl-2::GFP* expression in F1 offspring from parents exposed to *P. vranovensis* returned to normal levels when animals were allowed to develop on plates seeded with only *E. coli* HB101 for 24 h (Supplementary Fig. 6a). Interestingly, unlike adult animals, we found that *cysl-1* is robustly expressed in hypodermal cells in embryos and larva stage animals (Supplementary Figs. 5a, c and 6b, c). Collectively, these data indicate that the expression of at least *cysl-2* is significantly increased in response to *P. vranovensis* and that *cysl-1* and *cysl-2* are most robustly expressed in hypodermal cells in F1 embryos from parents exposed to *P. vranovensis*.

CYSL-1 and CYSL-2 were previously found to function in a signaling pathway with the regulator of hypoxia factor, RHY-1, the EGLN1 homolog EGL-9, and the hypoxia inducible factor HIF-1 to regulate animals' response to hypoxia[35]. We found that one of these genes, *rhy-1*, was also among the genes that exhibited the largest increase in expression in response to *P. vranovensis* (Fig. 5a) and was similarly upregulated in *wdr-23* mutant embryos (Supplementary Data 7). By contrast, we found that

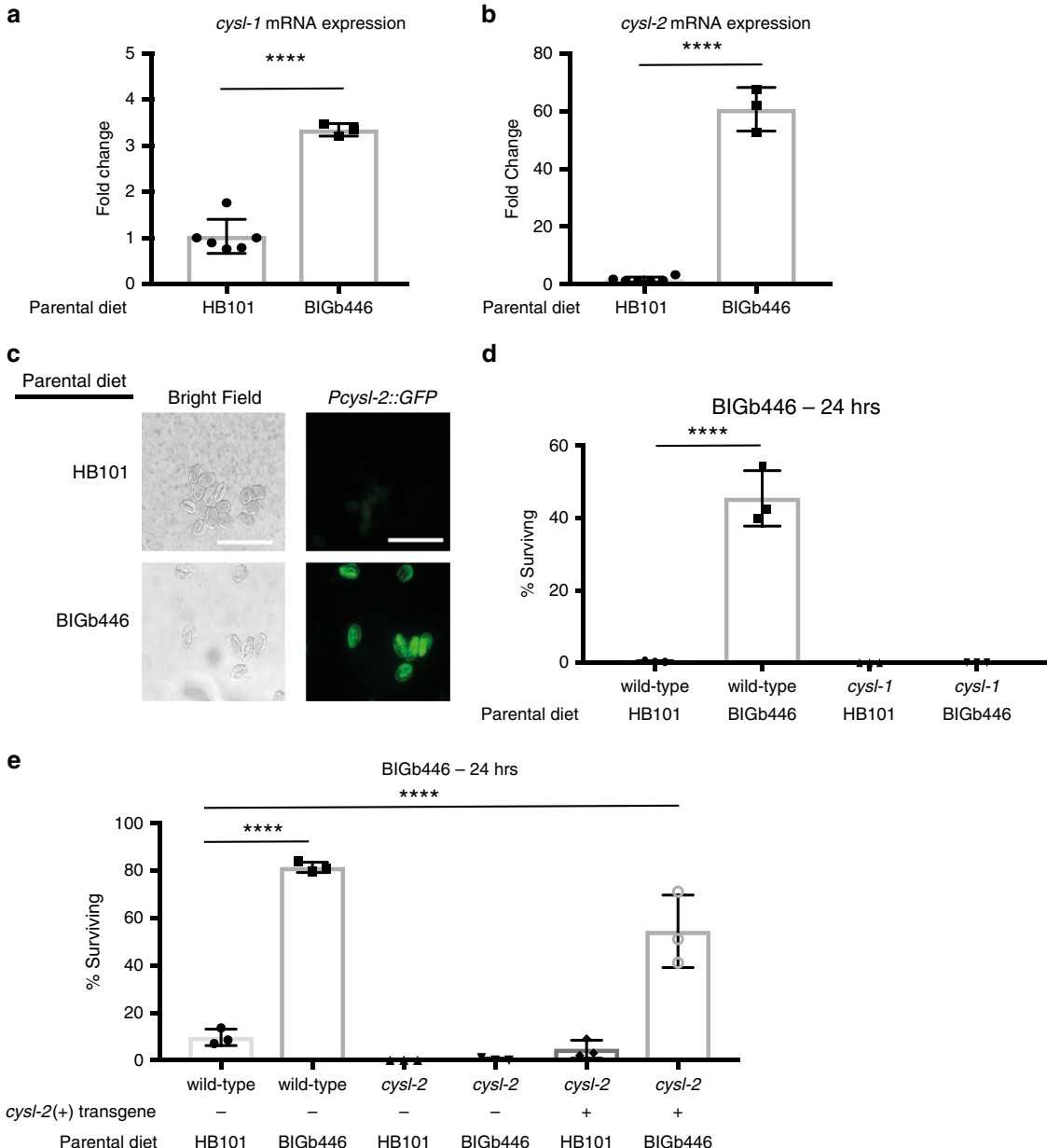

**Fig. 4 CYSL-1 and CYSL-2 are required for *C. elegans* to adapt to *P. vranovensis*. a** Fold Change of *cysl-1* mRNA in wild-type embryos from parents fed *E. coli* HB101 or *P. vranovensis* BIGb446. Data presented as mean values ± s.d. *n* = 3–6 replicates. **b** Fold Change of *cysl-2* mRNA in wild-type embryos from parents fed *E. coli* HB101 or *P. vranovensis* BIGb446. Data presented as mean values ± s.d. *n* = 3–6 replicates. Data is the same as found in Supplementary Data 5 and 7. **c** Representative images of *cysl-2*::GFP in embryos from parents fed *E. coli* HB101 or *P. vranovensis* BIGb446. Scale bars 100 μm. **d** Percent of wild-type and *cysl-1(ok762)* mutants surviving on plates seeded with bacterial isolates BIGb446 after 24 h. Data presented as mean values ± s.d. *n* = 3 experiments of >100 animals. **e** Percent of wild-type and *cysl-2(ok3516)* mutants surviving on plates seeded with bacterial isolates BIGb446 after 24 h. Data presented as mean values ± s.d. *n* = 3 experiments of >100 animals. ****$p < 0.0001$. Source data are provided as a Source Data file. See section "Statistics and reproducibility" for statistical tests run.

the expression of *egl-9* and *hif-1* was not regulated by either *P. vranovensis* or *wdr-23*. We tested whether mutants lacking RHY-1, EGL-9, and HIF-1 also exhibited altered adaptation to *P. vranovensis*. We found that three independent mutations in *rhy-1* resulted in animals that did not adapt to *P. vranovensis* (Fig. 5b), similar to *cysl-1* and *cysl-2* mutants. This defect was rescued by expressing a wild-type copy of *rhy-1* (Fig. 5b). By contrast, we found that the loss of *hif-1* and *egl-9* did not affect adaptation to *P. vranovensis* (Fig. 5c). These results are consistent with our observations that *cysl-1, cysl-2* and *rhy-1*, but not *egl-9* and *hif-1*, exhibit increased expression in response to

*P. vranovensis* and indicate that RHY-1, CYSL-1, and CYSL-2 function separately from EGL-9 and HIF-1 to promote animals' adaptation to *P. vranovensis*.

## Discussion

Our results demonstrate that *P. vranovensis* is a pathogen of *C. elegans* and that parental exposure of *C. elegans* to *P. vranovensis* can protect offspring from future infection via a mechanism that requires the cysteine synthases CYSL-1 and CYSL-2 and the regulator of hypoxia inducible factor RHY-1 (Fig. 6). The ability

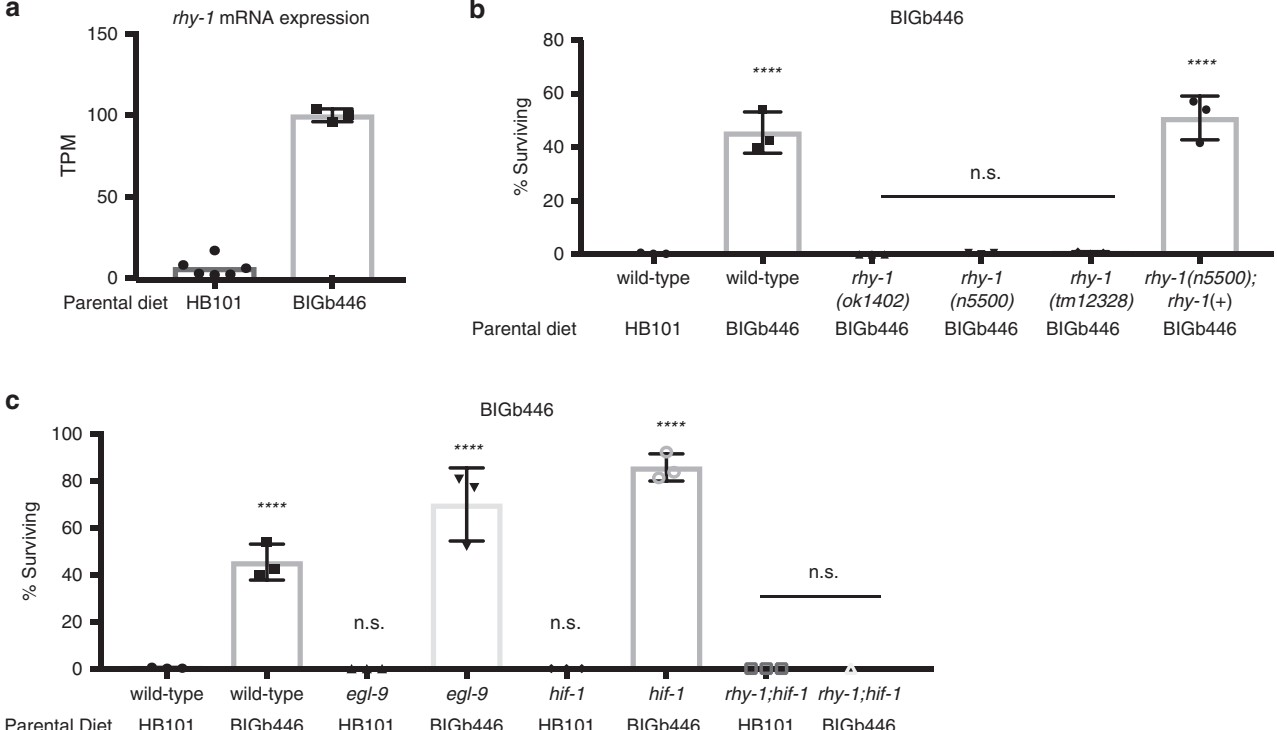

**Fig. 5 RHY-1 is required for *C. elegans* adaptation to *P. vranovensis*. a** Transcripts per million (TPM) of *rhy-1* in wild-type embryos from parents fed *E. coli* HB101 or *P. vraonvensis* BIGb446. Data presented as mean values ± s.d. *n* = 3–6 replicates. **b** Percent of wild-type and *rhy-1* mutants surviving on plates seeded with bacterial isolates BIGb446 after 24 h. Data presented as mean values ± s.d. *n* = 3 experiments of >100 animals. **c** Percent of wild-type, *egl-9(n586), hif-1 (ia4)*, and *rhy-1(n5500)* mutants surviving on plates seeded with bacterial isolates BIGb446 after 24 h. Data presented as mean values ± s.d. *n* = 3 experiments of >100 animals. ****$p$ < 0.0001. Source data are provided as a Source Data file. See section "Statistics and reproducibility" for statistical tests run.

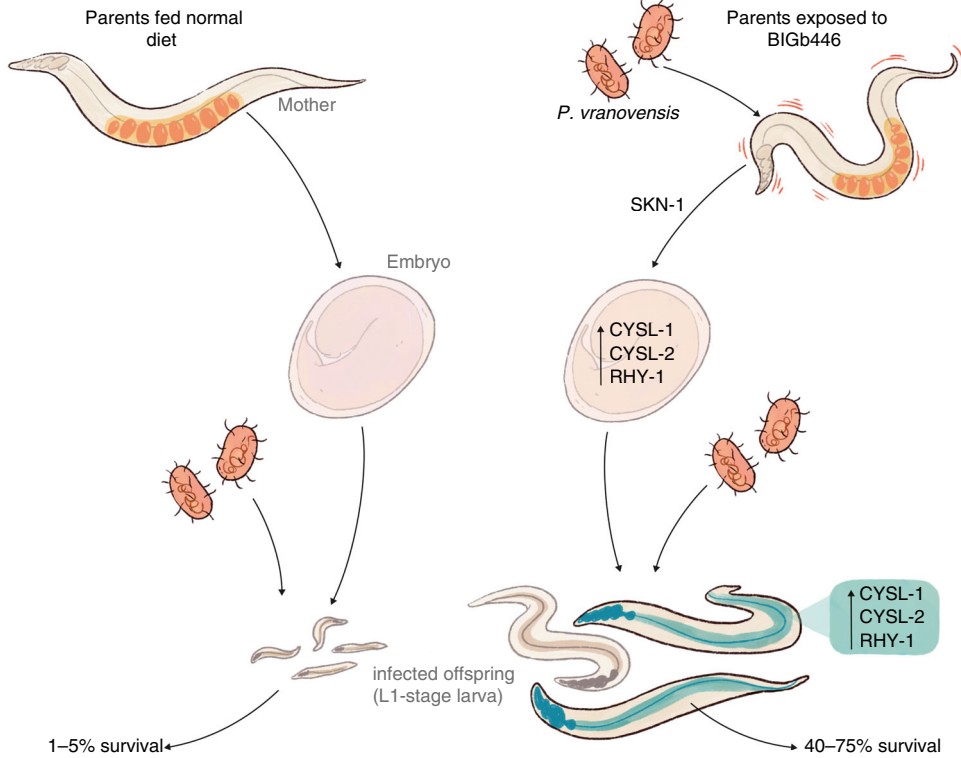

**Fig. 6 Model of *C. elegans* adaptation to *P. vranovensis*.** *C. elegans* heritably adapts to *P. vranovensis* exposure via a mechanism that requires CYSL-1, CYSL-2, and RHY-1. Briefly, our data indicates that parental exposure of *C. elegans* to *P. vranovensis* results in the activation of the SKN-1 transcription factor and increased expression of the cysteine synthases *cysl-1* and *cysl-2* and the predicted O-acyltransferase *rhy-1* in offspring. These three genes are required for the observed increase in offspring survival in response to *P. vranovensis* infection.

of *P. vranovensis* to kill *C. elegans* larvae within 24 h and the observation that *C. elegans* activates genes associated with oxidative stress in response to *P. vranovensis* suggests that *P. vranovensis* produces a toxic molecule(s) that is lethal to *C. elegans*. The cysteine synthases CYSL-1 and CYSL-2 have previously been reported to break down hydrogen cyanide[34]. In addition, RHY-1 was recently reported to function downstream of SKN-1 to promote resistance to hydrogen sulfide[36]. Similar to our observations, RHY-1 function downstream of SKN-1 was found to be independent of EGL-9/HIF-1 signaling[36]. Collectively, these observations suggest that *C. elegans* might heritably adapt to infection by *P. vranovensis* by increasing the expression of CYSL-1, CYSL-2, and RHY-1 in offspring, which in turn protects offspring by breaking down bacterially produced toxins such as hydrogen cyanide. However, we note that *rhy-1* mutants were previously reported to be resistant to hydrogen cyanide-mediated killing by *P. aeruginosa*[37]. By contrast, we found that *rhy-1* mutants are hypersensitive to *P. vranovensis* (Fig. 6b). These data suggest that resistance to potential hydrogen cyanide production by *P. vranovensis* alone does not explain the heritable adaptation to *P. vranovensis*.

Recent studies of *C. elegans* response to the human opportunistic pathogen *P. aeruginosa* have described a transgenerational response of animals to this pathogenic species of *Pseudomonas*[8]. These observations are phenotypically similar to our observations in that parental exposure to *P. aeruginosa* enhances F1 survival in response to future exposure to *P. aeruginosa*[8]. However, these observations are distinct from our observations in that our observed effect only lasts for one generation in most cases and the observed multigenerational effect only emerges after three consecutive generations of exposure to *P. vranovensis*. Studies of *C. elegans* response to *P. aeruginosa* found that exposure of a single generation of animals to *P. aeruginosa* resulted in a heritable increase in offspring survival that is mediated by a change in avoidance behavior that persisted for four generations. This heritable change in behavior was found to be dependent on the PIWI Argonaute homolog PRG-1[8]. By contrast, we found that *C. elegans* heritable adaptation to *P. vranovensis* did not require PRG-1 (Supplementary Fig. 3a) and was unlikely to be due to a change in avoidance behavior, because we used plates that were completely covered with a *P. vranovensis* lawn to prevent animals from avoiding the pathogen (Fig. 1a). In addition, we found that the heritable adaptation to *P. vranovensis* required SKN-1, CYSL-1, CYSL-2, and RHY-1 (Fig. 6). Based on these observations we propose that *C. elegans* heritable adaptation to *P. vranovensis* is both phenotypically and mechanistically distinct from *C. elegans* adaptation to *P. aeruginosa*.

Studies of *C. elegans* interaction with the opportunistic pathogen *P. aeruginosa* indicate that parental exposure to *P. aeruginosa* can have both adaptive[8] and deleterious[1] consequences for offspring. Similarly, studies of other heritable adaptive effects, such as *C. elegans* heritable adaptation to osmotic stress, have demonstrated that the adaptation to osmotic stress comes at the expense of animal's ability to survive anoxia and that this trade off might explain why the adaptive effect in offspring disappears shortly after adults are removed from osmotic stress[38]. We suspect that similar trade-offs, where adaptation to one stress comes at the cost of fitness in a different environment, are also likely to be the case for *C. elegans* adaptation to *P. vranovensis*. In addition, we hypothesize that potential costs of adapting might explain why the adaptation is lost if animals are removed from the pathogen for 24 h (Supplementary Fig. 2). Future studies will likely be critical in determining what the potential costs of heritable adaptations to stress are.

Finally, our results demonstrate that parental exposure to *P. vranovensis* resulted in the activation of *cysl-2*, and likely other

genes, only after gastrulation. The mechanisms by which this heritable change in gene expression are unclear, but there are several possibilities. For example, it is possible that parental exposure to *P. vranovensis* alters the chromatin landscape at pathogen response genes, such as *cysl-2* and this altered chromatin landscape explains the change in gene expression. An alternative explanation is that an altered form of the *skn-1* transcript or post-translationally modified SKN-1 protein is differentially deposited into embryos in a way that alters offspring gene expression. A third possibility is that a parental hormone, signal, metabolite, or toxin is differentially deposited into embryos in a way that alters gene expression after gastrulation. Future studies into the exact mechanism regulating gene expression in the offspring of parents exposed to *P. vranovensis* will significantly advance our understanding of how multigenerational responses to infection are mediated.

## Methods

**Strains**. *C. elegans* strains were cultured and maintained at 20 °C unless noted otherwise. The Bristol strain N2 was the wild-type strain.

 LGI: *set-32(ok1457), prg-1(n4357), spr-5(by134), wdr-23(mac32)*
 LGII: *cysl-2(ok3516, syb1431), damt-1(gk961032), rhy-1(n5500, ok1402, tm12398))*
 LGIII: *wdr-5.1(ok1417), hrde-1(tm1200), set-2(ok952), met-2(n4256)*
 LGIV: *pmk-1(km25), pgl-1(bn102), lin-45(n2018), nIs470[cysl-2::Venus; myo-2::RFP]; skn-1(zj15)*
 LGV: *egl-9(n586), hif-1(ia4)*
 LGX: *cysl-1(ok762, mr23, mr25, mr26, mr29, mr39, mr40)*
 Unknown linkage: *burIs2[gst-31::gfp]; nIs500 [cysl-1::gfp]*
 Extrachromosomal arrays: *nEx1763[rhy-1(+); myo-2::RFP]; burEx1 [cysl-2(+); myo-3::RFP];*

**Sequencing of *P. vranovensis* BIGb446 and BIGb468**. Genomic DNA was prepped using a Gentra Puregene kit (QIAGEN). DNA was sheared to 10 kb using gTUBE (Covaris). Sheared DNA was barcoded and multiplexed for PacBio sequencing using Template Prep Kit 1.0-SPv3 and Barcoded Adapter Kit 8A (PacBio). Genomic DNA was sequenced using the PacBio Sequel system using version 3.0 sequencing reagents (PacBio) and 1M v3 SMRT cell.

**Genome assembly of BIGb446 and BIGb468**. The genomes of BIGb446 and BIGb46 were assembled using HGAP4 from SMRT Link version 5.1.0.26412 with estimated genome sizes of 5 MB.

**Assays of adaptation to *P. vranovensis* BIGb446 and BIGb468**. *P. vranovensis* BIGb446 and BIGb468 was cultured in LB at 37 °C overnight. 1 mL of overnight culture was seeded onto 50 mm NGM agar plates and dried in a laminar flow hood (bacterial lawns completely covered the plate such that animals could not avoid the pathogen). All plates seeded with BIGb446 or BIGb468 were used the same day they were seeded. Young adult animals were placed onto 50 mm NGM agar plates seeded with 1 mL either *E. coli* HB101 or *P. vranovensis* BIGb446 or BIGb468 for 24 h at room temperature (22 °C). Embryos from these animals were collected by bleaching (unless otherwise stated) and placed onto fresh NGM agar plates seeded with BIGb446 or BIGb468. Percent surviving were counted after 24 h at room temperature (22 °C) unless otherwise noted. For Supplementary Fig. 1 embryos were transferred to new plates seeded with *P. vranovensis* by individually picking embryos.

**Multigenerational adaptation to *P. vranovensis***. *P. vranovensis* BIGb446 was cultured in LB at 37 °C overnight. 1 mL of overnight culture was seeded onto 50 mm NGM agar plates and dried in a laminar flow hood (bacterial lawns completely covered the plate). All plates seeded with BIGb446 were used the same day they were seeded. All animals in each generation were grown from embryos to young adults on NGM agar plates seeded with *E. coli* HB101. Young adults were then moved to fresh plates seeded with BIGb446 at room temperature (22 °C) for 24 h. 24 h of feeding on BIGb446 was counted as one generation of exposure to a BIGb446 diet. Embryos from parents fed BIGb446 were collected and placed onto plates seeded with *E. coli* HB101 until they were young adults and then moved to fresh plates seeded with BIGb446 for additional generations.

**Imaging of wild-type animals surviving after *P. vranovensis* exposure**. 1000 embryos were placed onto NGM agar plates seeded with *P. vranovensis* BIGb446 at $t = 0$ in each condition and surviving animals at 120 h were washed off of plates and resuspended in 20 μl M9 and imaged.

**Assays of *C. elegans* response to *P. aeruginosa* and *P. luminescens*.** Young adult animals were placed onto slow-killing assay (*P. aeruginosa*) plates or NGM agar (*P. luminescens*) plates seeded with either *P. aeruginosa* PA14 or *P. luminescens* Hb for 24 h at room temperature (22 °C). Embryos from these animals were collected and snap frozen in liquid nitrogen for RNA sequencing and metabolomics analysis or placed onto fresh NGM agar plates seeded with BIGb446. Percent of animals placed on BIGb446 surviving were counted after 24 h at room temperature (22 °C).

**Assay of adult survival.** Greater than 100 young adult animals were placed onto NGM agar plates seeded with *P. vranovensis* BIGb446 from overnight cultures in LB. Percent of animals surviving was counted at 24 h intervals. Animals were scored as alive if they were mobile and dead if they were immobile and did not respond to touch.

**Assays on carbenicillin plates.** L4 stage animals were placed onto NGM plates seeded with either *E. coli* HB101 or *P. vranovensis* BIGb446 for 24 h at room temperature (22 °C). Adult hermaphrodites and males were individually transferred to new plates containing carbenicillin at 25 μg/mL and seeded with carbenicillin resistant *E. coli* by picking and allowed to mate for 24 h. Embryos from these plates were then transferred to fresh plates seeded with *P. vranovensis* and the number of surviving animals was assayed after 24 h.

**RNA-seq.** Wild-type or *wdr-23(mac32)* mutant young adult animals were placed onto NGM agar plates seeded with *E. coli* HB101, *P. vranovensis* BIGb446 or BIGb468, or *P. luminescens* Hb or slow-killing assay plates seeded with *P. aeruginosa* PA14 for 24 h at room temperature (22 °C). Adult animals or embryos collected from adult animals after 24 h were snap frozen in liquid nitrogen. Samples were lysed using a BeadBug microtube homogenizer (Sigma) and 0.5 mm Zirconium beads (Sigma). RNA was extracted using a RNeasy Plus Mini kit (Qiagen). mRNA was enriched using an NEBNext rRNA Depletion kit (NEB). Libraries for sequencing were prepared using an NEBNext Ultra II Library prep kit for Illumina (NEB) and loaded for paired-end sequencing using the Illumina HiSeq 1500.

**GFP imaging.** To image embryos and adults, L4-stage animals expressing either *nIs470*, *nIs500*, or *burIs2* were placed onto NGM agar plates seeded with either *E. coli* HB101 or *P. vranovensis* BIGb446 at room temperature (22 °C) for 24 h. Embryos and adults in Fig. 4c and Supplementary Fig. 4 were collected and immediately imaged using a Zeiss AXIO imager A1 microscope and a Hamamatsu ORCA-ER camera. Embryos and adults in Supplementary Figs. 3 and 4 were imaged using a Leica DM6 B and a Leica DFC9000 GT camera. To image L2-stage larva in Supplementary Fig. 4, embryos collected from parents exposed to *P. vranovensis* were allowed to develop on plates seeded with *E. coli* HB101 for an additional 24 h at room temperature (22 °C) and then imaged using a Leica DM6 B and a Leica DFC9000 GT camera.

**UV-killing bacteria.** 1 mL of overnight culture of BIGb446 was seeded onto 50 mm NGM agar plates and dried in the laminar flow hood. Seeded plates were then exposed to 20 μW/cm$^2$ for 1 h. Complete killing of bacteria was confirmed by testing inoculations of bacteria in LB overnight.

**cysl-2 cloning and rescue.** Genomic *cysl-2* DNA was amplified from wild-type animals using the *cysl-2* fwd primer ACGATTGGGTTGGCTGTAAG and the *cysl-2* rev primer GGTCGTACGTGTTCGTTGTG. Extrachromosomal arrays were generated by injecting the corresponding PCR fragment and co-injection marker into the gonad of one-day old adults at the specified concentrations. *nobEx1* was generated by injecting *cysl-2* genomic DNA at 20 ng/μl and *myo-3::RFP* was injected at 10 ng/μl. Final injection DNA concentration was brought up to 150 ng/μl using DNA ladder (1 kb HyperLadder—Bioline).

**Generation of *cysl-2* CRISPR alleles.** *syb1431* was generated by SunyBiotech. *syb1431* contains a 50 bp frameshift deletion in the first exon with the following flanking sequences ACCGGTGGTGAGCTCATCGGAAACACCCCA and GGTAGAGTACATGAACCCTGCCTGCTC.

**LC/MS lipid profiling.** Approximately, 40 μL of concentrated embryos were re-suspended in 100 μL of water, then 0.4 mL of chloroform was added to each sample followed by 0.2 mL of methanol containing the stable isotope labeled acyl-carnitine internal standards (Butyryl-L-carnitine-$_{d7}$ at 5 μM and Hexadecanoyl-L-carnitine-$_{d3}$ at 5 μM). The samples were then homogenized by vortexing then transferred into a 2 mL Eppendorf screw-cap tube. The original container was washed out with 0.5 mL of chloroform:methanol (2:1, respectively) and added to the appropriate 2 mL Eppendorf screw-cap tube. This was followed by the addition of 150 μL of the following stable isotope-labeled internal standards (~10–50 μM in methanol): Ceramide_$C16_{d31}$, LPC_$(C14:0_{d42})$, PC_$(C16:0_{d31}/C18:1)$, PE_$(C16:0_{d31}/C18:1)$, PG_$(C16:0_{d31}/C18:1)$, PI_$(C16:0_{d31}/C18:1)$, PS_$(C16:0_{d62})$, SM_$(C16:0_{d31})$, TG_

(45:0_$_{d29}$), and TG_(48:0_$_{d31}$). Then, 400 μL of sterile water was added. The samples were vortexed for 1 min, and then centrifuged at ~20,000 rpm for 5 min.

For the intact lipid sample preparation, 0.3 mL of the organic layer (the lower chloroform layer) was collected into a 2 mL amber glass vial (Agilent Technologies, Santa Clara, CA, USA) and dried down to dryness in an Eppendorf Concentrator Plus system (Eppendorf, Stevenage, UK) run for 60 min at 45 °C. The dried lipid samples were then reconstituted with 100 μL of 2:1:1 solution of propan-2-ol, acetonitrile, and water, respectively, and then vortexed thoroughly. The lipid samples were then transferred into a 300 μL low-volume vial insert inside a 2 mL amber glass auto-sample vial ready for liquid chromatography separation with mass spectrometry detection (LC–MS) of intact lipid species.

For the acyl-carnitine sample preparation, 0.2 mL of the organic layer (the lower chloroform layer) and 0.2 mL of the aqueous layer (the top water layer) were mixed into a 2 mL amber glass vial and dried down to dryness. The dried acyl-carnitine samples were then reconstituted with 100 μL of water and acetonitrile (4:1, respectively) and thoroughly vortexed. The acyl-carnitine samples were then transferred into a 300 μL low-volume vial insert inside a 2 mL amber glass auto-sample vial ready for liquid chromatography separation with mass spectrometry detection (LC–MS) of the acyl-carnitine species.

Full chromatographic separation of intact lipids[39] was achieved using Shimadzu HPLC System (Shimadzu UK Limited, Milton Keynes, UK) with the injection of 10 μL onto a Waters Acquity UPLC® CSH C18 column; 1.7 μm, I.D. 2.1 mm × 50 mm, maintained at 55 °C. Mobile phase A was 6:4, acetonitrile and water with 10 mM ammonium formate. Mobile phase B was 9:1, propan-2-ol and acetonitrile with 10 mM ammonium formate. The flow was maintained at 500 μL per minute through the following gradient: 0.00 min_40% mobile phase B; 0.40 min_43% mobile phase B; 0.45 min_50% mobile phase B; 2.40 min_54% mobile phase B; 2.45 min_70% mobile phase B; 7.00 min_99% mobile phase B; 8.00 min_99% mobile phase B; 8.3 min_40% mobile phase B; 10 min_40% mobile phase B; 10.00 min_40% mobile phase B. The sample injection needle was washed using 9:1, 2-propan-2-ol, and acetonitrile with 0.1% formic acid. The mass spectrometer used was the Thermo Scientific Exactive Orbitrap with a heated electrospray ionization source (Thermo Fisher Scientific, Hemel Hempstead, UK). The mass spectrometer was calibrated immediately before sample analysis using positive and negative ionization calibration solutions (recommended by Thermo Scientific). Additionally, the heated electrospray ionization source was optimized at 50:50 mobile phase A to mobile phase B for spray stability (capillary temperature; 380 °C, source heater temperature; 420 °C, sheath gas flow; 60 (arbitrary), auxiliary gas flow; 20 (arbitrary), sweep gas; 5 (arbitrary), source voltage; 3.5 kV. The mass spectrometer resolution was set to 25,000 with a full-scan range of *m/z* 100–1800 Da, with continuous switching between positive and negative modes. Lipid quantification was achieved using the area under the curve (AUC) of the corresponding high resolution extracted ion chromatogram (with a window of ±8 ppm) at the indicative retention time. The lipid analyte AUC relative to the associated internal standard AUC for that lipid class was used to semi-quantify and correct for any extraction/instrument variation.

Acyl-carnitine chromatographic separation was achieved using an ACE Excel 2 C18-PFP (150 mm, I.D. 2.1 mm, 2 μm) LC-column with a Shimadzu UPLC system. The column was maintained at 55 °C with a flow rate of 0.5 mL/min. A binary mobile phase system was used with mobile phase A; water (with 0.1% formic acid), and mobile phase B; acetonitrile (with 0.1% formic acid). The gradient profile was as follows; at 0 min_0% mobile phase B, at 0.5 min_100% mobile phase B, at 5.5 min_100% mobile phase B, at 5.51 min_0% mobiles phase B, at 7 min_0% mobile phase B. Mass spectrometry detection was performed on a Thermo Exactive orbitrap mass spectrometer operating in positive ion mode. Heated electrospray source was used; the sheath gas was set to 40 (arbitrary units), the aux gas set to 15 (arbitrary units) and the capillary temperature set to 250 °C. The instrument was operated in full scan mode from *m/z* 75–1000 Da. Acyl-carnitine quantification was achieved using the area under the curve (AUC) of the corresponding high resolution extracted ion chromatogram (with a window of ±8 ppm) at the indicative retention time. The acyl-carnitine analyte AUC relative to the associated internal standard AUC was used to semi-quantify and correct for any extraction/instrument variation. All lipid values were normalized to total lipid.

**RNA-seq and principal component analysis.** Cutadapt version 1.18 was used to remove adapter sequences (AATGATACGGCGACCACCGAGATCTACACTCTT TCCCTACACGACGCTCTTCCGATC for HS678 and AGATCGGAAGAGCACA CGTCTGAACTCCAGTCA (forward) and AGATCGGAAGAGCGTCGTGTAGG GAAAGAGTGT (reverse) for HS755. Cutadapt was also used to trim the 3′ ends of reads when the phred quality value was below 20. Reads shorter than 40 bp were discarded. The genome sequence in FASTA format and annotation file in GFF3 format of *C. elegans* were downloaded from Ensembl release 96. The genome sequence was indexed with the annotation with hisat2 version 2.1.0. hisat2 was used for aligning reads to the reference genome and the maximum number of alignments to be reported per read was set to 5000. Feature counts from the conda subread package version 1.6.3 was used to count the number of reads per gene. The Ensembl release 96 annotation file in GTF format for *C. elegans* was used. Fragments were counted when they overlapped an exon by at least 1 nucleotide and fragments are reported at the gene level. The option for the stranded protocol was

turned off. Only read pairs that had both ends aligned were counted. Given our average fragment length of 300 bp, a distance of between 50 and 600 nucleotides was tolerated for read pairs. Read pairs that had their two ends mapping to different chromosomes or mapping to the same chromosome but on different strands were not counted. Multi-mapping reads were not counted. samtools (https://www.ncbi.nlm.nih.gov/pubmed/19505943) was used for SAM/BAM format conversion and BAM indexing. The raw counts table was imported into R 3.5.1 for differential expression analysis with DESeq2 version 1.22.1. Normalization was carried out within each contrast. A snakemake workflow[40] was created for the RNA-seq analysis and can it be found at https://github.com/cristianriccio/celegans-pathogen-adaptation.

**Genomic DNA alignment**. MUMmer (Version 3.0, default parameters) and MUMmer-plot[41] were used to visualize global alignments of BIGb446 and BIGb468 whole genome assemblies. Further to this NUCmer[41] and the associated dnadiff script was used to produce statistics on the alignment of the two bacterial genomes.

**Statistics and reproducibility**. One-way ANOVA analysis with post hoc $p$-value calculations was used for Figs. 1b, e, 2a, b, 3c, 4d, e, 5b, c and Supplementary Figs. 3a, c, and 4c. Two-tail $t$-tests were used for Figs. 1c, 3f, 4a, b, 5a, and Supplementary Fig. 3b. For Fig. 3b, d, e the $p$-value was calculated using a normal approximation to the hypergeometric probability, as in http://nemates.org/MA/progs/representation.stats.html. For lipidomics data (Supplementary Fig. 3 and Supplementary Data 4) $p$-values were calculated using two-tail $t$-tests and statistical significance was determined using Bonferonni correction for multiple hypotheses. $*p < 0.05$, $**p < 0.01$, $***p < 0.001$, and $****p < 0.0001$. No statistical method was used to predetermine sample size. Sample sizes were chosen based on similar studies in the relevant literature. The experiments were not randomized. The investigators were not blinded to allocation during experiments and outcome assessment. Source data are provided as a Source Data file.

**Reporting summary**. Further information on research design is available in the Nature Research Reporting Summary linked to this article.

## Data availability

RNA-seq data that support the findings of this study have been deposited at the European Nucleotide Archive (ENA) under the accession code PRJEB32993. The raw data for assembling the genomes of *P. vranovensis* isolate BIGb446 is available at the ENA under the accession code ERS3670403 and BIGb468 is available under the accession code ERS3670404. The raw data related to metabolomics is available on Metabolomics Workbench using project code PR000904. The source data underlying Figs. 1b, c, e, 2a, b, 3c, f, 4a, d, e, 5a, b, c and Supplementary Figs. 2c, 3a, b, c, and 4c are provided as a Source Data file.

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

## Acknowledgements

We thank Buck Samuel, Gary Ruvkun, and Jonathan Ewbank for bacterial isolates; Long Ma, Keith Choe, Derek Sieburth, Peter Urwin, Bob Horvitz, Na An, and the *Caenorhabditis* Genetic Center, which is funded by the NIH National Center for Research Resources (NCRR), for strains, Martin Welch and George Salmond for use of bacterial culture equipment and space, Martin Hemberg for feedback on RNA-seq analysis, and Claudia Flandoli for help in designing models. N.O.B. is funded by a Next Generation Fellowship from the Centre for Trophoblast Research. C.R. was supported by a Wellcome Ph.D. program. A.K. and B.J. are funded by BBSRC grant BB/M027252/1. This work was also supported by Cancer Research UK (C13474/A18583, C6946/A14492) and the Wellcome Trust (104640/Z/14/Z, 092096/Z/10/Z) grants to E.A.M.

## Author contributions

N.O.B. conceived the project and designed the experiments. N.O.B. and E.A.M. analyzed the data. N.O.B., C.R., A.D., J.P., A.K., and B.J. performed the experiments. N.O.B. wrote the manuscript.

## Competing interests

The authors declare no competing interests.
