## [Peer Review File · Nature Communications]

Reviewers' comments:

Reviewer #1 (Remarks to the Author):

In this manuscript Burton et al. report that the exposure of adult *C. elegans* to the pathogen *Pseudomonas vranovensis* allows their embryonic progeny to survive to the exposure to the same pathogen. They have also demonstrated the requirement of three genes, the cysteine synthases *cysl-1* and *cysl-2* and the regulator of hypoxia inducible factor *rhy-1* in this adaptive pathogen resistance response. Overall the experiments shown are well conducted, even though some of the data can be presented better. Given the fact that most of the transgenerational effects observed to date are very modest, the authors did a great job in identifying such a robust phenomenon of adaptation. However, the lack of a mechanistic understanding of the process is a limitation of the study and more experiments are needed to reinforce their conclusions. Also, they are observing more likely a maternal effect rather than a real transgenerational adaptive response.

Specific comments:

A. The experiment shown in Figure 1 are nice in showing the survival of embryos grown on pathogenic food upon previous exposure to the pathogen in the adults. However, there is lack of explanatory information (in the main text, method, and figure) on how the experiment is executed. For instance, a graphical representation of the assay might help the readers to better understand how the experiment is conducted, instead of showing the panel C. One of my concerns is that after the exposure of a young adult worm for 24h there must be a mixture of hatched larvae and embryos at different stages of development. Therefore, the mixed stage embryos collected are deriving from an adult that has been exposed to the pathogen for different time. Maybe they can try to collect embryos all at similar developmental stage and/or assay embryos that are laid earlier or later to see whether they show different survival responses to the pathogen.

B. Can the authors perform a similar assay on adult F1 worms in addition to the assay with embryos? Do they have the same results? Also, they should perform a similar experiment using males (with a fluorescent reporter) exposed to pathogen and test whether they are able to transmit to the embryos of cross-fertilized hermaphroditic animal (without exposure to pathogen) the adaptive response. This experiment will clarify if this is a maternal effect.

C. In Figure 2, the authors claim that *C. elegans* exposure to *P. vranovensis* can have transgenerational effects on progeny survival. I am not sure they are showing here a real transgenerational effect. First, the survival after exposure for 24h in adulthood for 3 generations is only shown to be significant in non-exposed F2 embryos. They should show the resistance in F2 adults and also for subsequent generations. If this is due to a maternal effect, it is possible that the three generations' exposure just increases the amount of inherited maternal product that will persist in the embryos of non-exposed animals. Again, a time-course with embryos laid at different timepoint might help in the interpretation of the results. Also, a better graphical representation of

the experiment might be good to have it here (it is a bit confusing for how long the adults in each generation have been exposed to the pathogen - continuously or only for 24h in YA stage).

D. In figure 5C the authors show a transcriptional reporter of CYSL-2. This is a nice tool although a CRISPR-Cas9 tagged version of CYSL-2 (or RHY-1) can be used to monitor the persistence of the protein in the embryo and its persistence even in F1 adults after the 3 generations exposure. This will allow to monitor whether the maternal induced proteins can persist until the next generation. Also, it would be interesting to perform RNA FISH experiments to detect *rhy-1*, *cysl-1* and *cysl-2* and follow the accumulation of these RNAs in adult tissues and embryos.

E. The changes in gene expression in embryos upon adult exposure to *P. vranovensis* are really large compared to adult exposure to *P. aeruginosa* and *P. luminescence*. Yet, mutation in only one gene is sufficient to abolish this effect and its transgenic expression is able to rescue the phenotype. Are all those detected gene expression changes due to developmental differences in mixed stages embryo collected (most of the changed genes are very low expressed)? To reinforce their data the authors should perform RNA-seq in *rhy-1* mutant embryos exposed to the pathogen (Fig. 6b) and check how many of the upregulated genes are specific to the *rhy-1* gene activity. Also, the RNA-seq in the rescued embryos (*rhy-1* (n5500); *rhy-1* +) can further confirm the specific gene expression changes.

F. Is the colonization of adult worms by *P. vranovensis* required for the adaptive response to *P. vranovensis* in embryos? Maybe the author should check the colonization level of adult worms exposed to fluorescent *P. vranovensis* (using bacterial GFP reporter for example). In addition, because RHY-1 is a hypoxia inducible factor, the authors can try to expose the adult worms to hypoxia condition to see whether this is sufficient to trigger the adaptive responses to *P. vranovensis* in the embryos.

Reviewer #2 (Remarks to the Author):

Burton et al. investigate the transgenerational heritability of pathogen defense response mechanisms in *C. elegans*. Recently, a number of publications have reported that *C. elegans* responds to the presence of pathogens by epigenetically regulating germline gene expression and that those epigenetic changes are inherited up to three consecutive generations. The topic of heritability of adaptive responses is highly interesting.

Burton et al. exposed worms to *P. vranovensis* and observe that the sensitivity of the animals is alleviated when the parental generation had been exposed to the pathogen. The authors should more clearly explain how those experiments were performed. They state in the methods part that embryos were collected. Here it is important to clarify whether bacterial contamination was excluded. Were the embryos collected by bleaching or just transferred by picking?

The authors report that culturing for multiple generations offers most protection and that a significant protection could be maintained to some degree when the last generation was exposed to non-pathogens.

They then exclude known epigenetic regulators and also *pmk-1* immune signaling as mediators of the transgenerational protection and instead perform gene expression analysis to identify genes responding to *P. vranovensis*. Here, they show that *cysl-1* and *cysl-2* as well as *rhy-1* are induced upon infection and required for the protection of offspring. The expression of those genes should be more thoroughly characterized. Where and when are they expressed? Only in embryos or in the parents? In which tissues/cell types? When are they switched off, are they *pmk-1* dependent?

The experiments appear to be well performed and the authors' conclusions are backed by the data. They identify with *cysl-1/-2* and *rhy-1* a potentially new mechanism of transgenerational pathogen resistance that appears to be distinct from the recently established epigenetic mechanisms. It remains unclear how those cysteine synthases and regulator of hypoxia inducible factor *RHY-1* operate in this context. No mechanistic insight into their biochemical or regulatory function and the phenotypic consequence is probed for. How is cysteine metabolism related to this? Is there any role for hypoxia? Is this a general stress response or a specific pathogen response?

Reviewer #3 (Remarks to the Author):

Burton et al - *C. elegans* heritably adapts to *P. vranovensis* infection via a mechanism that requires the cysteine synthases *CYSL-1* and *CYSL-2* ‘

Here the authors examine the effects of the soil bacteria *P. vranovensis*, which they identified as a natural bacterial isolate, on *C. elegans* and its offspring. They find that a single generation of exposure to *Pv* did not induce the multigenerational effect, but rather only acts intergenerationally, unless treated consecutively from P0 through F3, the F5 generation survived longer. They find that many genes involved in transgenerational inheritance from a short exposure to bacteria are not involved; instead, *PMK-1* is required for survival after *Pv* exposure, but not for the adaptation effect. Other pathogens did not induce adaptation to *Pv* (see note below). Gene expression and metabolic profiling were carried out. The cysteine synthases *CYSL-1* and *-2* are increased in embryos exposed to *Pv*, and mutants in *cysl-1* and *cysl-2* are unable to adapt to *Pv*; similarly *rhy-1* mutants did not adapt, but *hf-1* and *egl-9* mutants do, suggesting the pathways are separate.

The authors discuss differences with *Pseudomonas aeruginosa* avoidance, which is transgenerational, vs offspring survival on *Pv* after 4 generations of exposure, which is more similar to dauer induction after P0-F2 exposure to *PAO1* and *S. enterica*.

This paper is somewhat inconclusive, in that it points out some differences with other paradigms, but doesn't clearly identify a distinguishing pathway or signaling mechanisms. It does have some strengths, however (interesting effect, clearly different from transgenerational *Pa* avoidance,

similarities to dauer induction after multigenerational exposure) so it would be interesting to see what the authors might be able to further flesh out mechanistically.

Major comments:

1. It seems like an exaggeration to write “These results indicate that the environment of P0 animals can enhance the survival of F4 generation” since it is really more correct to write P0 through F2 (three generations)- it is not the P0 environment, but rather that plus the next two generations of exposure that induces the observed increase in offspring survival.

2. The logic of exposing the worms to two other pathogens, then testing offspring survival to Pv seems turned around. First, the authors should test whether the other pathogens induce a similar effect as Pv does (i.e., does P0-F2 exposure to PA14 increase offspring survival in the F4 generation exposed to PA14?), and whether multigenerational Pv exposure induces offspring survival to the other pathogens – these will better answer the question posed: whether heritable adaptation to Pv is a general or specific response.

3. Was anything learned from the gene expression profiling that is described before the metabolomics analysis? This seems like a lost opportunity.

4. The study does not identify the reason (underlying mechanism) for the differences in adaptation to Pv vs *Pseudomonas aeruginosa*.

5. In the Discussion, the authors compare the differences in responses to Pv and Pa, but the assays used for the two are different (avoidance to Pa vs offspring survival on Pv), so it is difficult to even make these comparisons.

6. No clear mechanistic model emerges, just the involvement of two cysteine synthases and RHY-1, but these roles are not well described either, beyond being required.

Minor comments:

1. The authors jump from describing BIGb446 and 468 to discussing *P. vranovensis* before the strains were identified as Pv (p. 5, line 137).

Reviewers' comments:

Reviewer #1 (Remarks to the Author):

In this manuscript Burton et al. report that the exposure of adult *C. elegans* to the pathogen *Pseudomonas vranovensis* allows their embryonic progeny to survive to the exposure to the same pathogen. They have also demonstrated the requirement of three genes, the cysteine synthases *cysl-1* and *cysl-2* and the regulator of hypoxia inducible factor *rhy-1* in this adaptive pathogen resistance response. Overall the experiments shown are well conducted, even though some of the data can be presented better. Given the fact that most of the transgenerational effects observed to date are very modest, the authors did a great job in identifying such a robust phenomenon of adaptation. However, the lack of a mechanistic understanding of the process is a limitation of the study and more experiments are needed to reinforce their conclusions. Also, they are observing more likely a maternal effect rather than a real transgenerational adaptive response.

We thank Reviewer 1 for their comments on the experiments presented here and agree that we are excited to find such a robust phenomenon of adaptation given the modest effects that make up much of the field of intergenerational and transgenerational effects. We have now added substantial new data both offering mechanistic insight and reinforcing our conclusions. We have also added further clarifying text and discussion of how our studies fit within the broader literature of maternal, intergenerational, multigenerational, and transgenerational effects and the overlap between these terms – see responses 1.4 and 1.5.

Specific comments:

A. The experiment shown in Figure 1 are nice in showing the survival of embryos grown on pathogenic food upon previous exposure to the pathogen in the adults. However, there is lack of explanatory information (in the main text, method, and figure) on how the experiment is executed. For instance, a graphical representation of the assay might help the readers to better understand how the experiment is conducted, instead of showing the panel C.

1.1 - We thank Reviewer 1 for this comment and agree that graphical representation of the assay will substantially help readers understand how the experiment is conducted. We have added such a graphical representation to the manuscript as Figure 1a.

One of my concerns is that after the exposure of a young adult worm for 24h there must be a mixture of hatched larvae and embryos at different stages of development. Therefore, the mixed stage embryos collected are deriving from an adult that has been exposed to the pathogen for different time. Maybe they can try to collect embryos all at similar developmental stage and/or assay embryos that are laid earlier or later to see whether they show different survival responses to the pathogen.

1.2 - We collected embryos by bleaching at 24 hours, so there are no hatched larvae collected in our assays. However, Reviewer 1 correctly points out that the collected embryos are mixed stage. To determine if differences in embryo staging underlie our observed adaptation, we have added new data to the manuscript where we transferred only 3-fold stage embryos by picking individual embryos. We found that the adaptation when transferring only 3-fold stage embryos is similar to our observations for mixed stage embryos collected by bleaching, indicating that differences in embryos staging do not underlie our observed adaptive effect (New Supplementary Fig. 2b and new lines 175-183 of the text).

B. Can the authors perform a similar assay on adult F1 worms in addition to the assay with embryos? Do they have the same results?

1.3 - We performed this assay and found F1 adults from parents exposed to *P. vranovensis* do not exhibit a statistically significant difference in survival when compared to F1 adults from parents fed only *E. coli* HB101. We have added this data to the manuscript and a discussion of it to the discussion section.

Briefly, there are several possible reasons for why F1 larvae exhibit an adaptation while F1 adults do not. One possibility is that the adaptive effect fades away if animals are removed from *P. vranovensis* for multiple days, as is the case in this assay of adults (embryo to L4 stage F1 animals develop feeding on only *E. coli*). This possibility is supported by new data we have added to the manuscript which demonstrates that treating adults with antibiotics for 24 hours eliminated the observed heritable adaptive effect (New Supplementary Fig. 1b), and would be similar to other heritable adaptations observed in *C. elegans* such as the maternally regulated heritable adaptation to osmotic stress (Frazier and Roth, 2011 and Burton et al., 2017) which is also eliminated shortly after adults are removed from osmotic stress.

A second possible explanation is that adults exhibit significantly reduced expression of *cysl-1* in hypodermal cells when compared to larvae and that differences in *cysl-1* expression in hypodermal cells underlies why larvae can adapt but adults do not. (New Supplemental Figs. 5 and 6). Specifically, we found that larvae express CYSL-1 predominantly in hypoderm and neuronal cells. By contrast, adults lose most of the observed hypodermal expression and exhibit CYSL-1 expression mostly in neurons. Given our findings that CYSL-1 is required for adaptation, this differing expression pattern of a key factor mediating adaptation might explain the observed difference between F1 larvae and F1 adults.

Finally, our data indicate that larvae die within hours of exposure to *P. vranovensis*, suggesting that they die due to some sort of toxin, whereas adults survive for multiple days before dying. It is possible that the heritable adaptive effect protects larvae from a toxin-mediated killing but cannot protect them from the slower killing observed in adults which might occur via a different mechanism analogous to the different mechanisms that mediate slow-killing and fast-killing when *C. elegans* are infected with *P. aeruginosa*.

Also, they should perform a similar experiment using males (with a fluorescent reporter) exposed to pathogen and test whether they are able to transmit to the embryos of cross-fertilized hermaphroditic animal (without exposure to pathogen) the adaptive response. This experiment will clarify if this is a maternal effect.

1.4 – We agree with Reviewer 1 that this would be a very informative experiment and we indeed attempted this experiment by taking either infected hermaphrodites or males and transferring them to new plates without pathogen to cross them with either uninfected males or uninfected hermaphrodites respectively. These plates contained antibiotics so that the infected animals did not simply transfer the infection to the uninfected animals. (Mating infected animals with uninfected animals in the absence of antibiotics resulted in both animals exhibiting pathogen response gene expression based on GFP reporters, likely due to uninfected animals becoming infected, which complicated any interpretation).

We found that removing hermaphrodites from *P. vranovensis* seeded plates, in order to perform the cross, and transferring them to plates containing antibiotics for 24 hours eliminated the adaptive effect in offspring, even in self-fertilized offspring from infected hermaphrodites. This result indicates that removing animals from the pathogen also eliminates the adaptive effect in offspring and has prevented us from interpreting the proposed experimental crosses. This data has now been added to the manuscript (New Supplemental Fig. 1b).

We note that this observation is similar to our previous studies of a heritable adaptive response to osmotic stress (Burton et al., 2017) where removing hermaphrodites from osmotic stress for 24 hours eliminates the adaptive effect observed in offspring. In the case of osmotic stress, the adaptation comes at the expense of offspring ability to survive anoxia, and thus if the osmotic stress goes away then animals might not want to continue to program an adaptation in offspring. It is possible that adaptation to *P. vranovensis* also comes at a cost and therefore parents would not want to continue to program this adaptation in offspring if the threat of infection is gone.

Thus, we added text to the results and discussion section discussing how our observations might represent a normally maternally regulated adaptation (similar to osmotic stress adaptation that is regulated by insulin-like signalling to oocytes) that persists for multiple generations under certain circumstances (lines 218-238 in the results section and lines 456-481 in the discussion).

We have also changed the word transgenerational to multigenerational in most cases in order to avoid causing any confusion among readers, but we note that that the multigenerational/transgenerational effect observed in Figure 2 could not properly be described as a maternal effect or an intergenerational effect (see response 1.5). Finally, we have added additional discussion of these terms and how our observations fit within them given the sometimes varied definitions of these terms used in the literature. We do not believe that such a change in wording affects how we think about the biology identified in this manuscript, the novelty of our findings, or the excitement about or importance of our findings to the broader field of multigenerational effects, which encompasses both intergenerational and transgenerational effects.

C. In Figure 2, the authors claim that *C. elegans* exposure to *P. vranovensis* can have transgenerational effects on progeny survival. I am not sure they are showing here a real transgenerational effect. First, the survival after exposure for 24h in adulthood for 3 generations is only shown to be significant in non-exposed F2 embryos. They should show the resistance in F2 adults and also for subsequent generations. If this is due to a maternal effect, it is possible that the three generations' exposure just increases the amount of inherited maternal product that will persist in the embryos of non-exposed animals.

1.5 - We agree with Reviewer 1 and think that the most likely explanation for our observed multigenerational/transgenerational effect on animal survival is that an inherited parental (likely maternal) signal builds up, changes, or is maintained differently upon three consecutive generations of exposure to *P. vranovensis* and this allows an effect that normally only lasts one generation to instead persist for two generations. We note, however, that it is very improbable that our observed effect of enhanced F4 progeny survival reported in Figure 2 is due to the simple build up and persistence of a randomly diluting maternal factor, and it is likely that this effect would require an active process to occur – see below. We believe that

this was not clear in our original text and we have therefore substantially revised this section of the text.

Briefly, for any inherited maternal product from P0 animals' environment to have a phenotypic effect on F4 animals (as we report here) it would have to persist through an estimated 10^8 -fold dilution at a minimum. This is because a single *C. elegans* hermaphrodite produces hundreds of progeny that go through a series of cell divisions before forming mature germ cells/producing new embryos. For the purposes of considering transgenerational effects in *C. elegans* it has been proposed that there is a minimum of 100-fold dilution between any one generation and the next (Schott et al., 2014) – this is a conservative estimate and the dilution is likely larger.

Fitting this conservative estimate with our data indicates that the exposure of three consecutive generations of animals in a row to any given stress can only contribute 0.01% more total product to F4 offspring, assuming random dilution, than the exposure of two generations of animals in a row, which we found to have no effect on F4 progeny survival.

Example -- if P0 exposed animals contribute 100 hypothetical units of maternal product to F1 animals when P0s are exposed to pathogen, then re-exposure of F1 animals to pathogen contributes a new 100 units to F2 offspring plus 1/100, after dilution, of the units it got from its P0 mother for a total of 101 units transferred to F2 animals. Re-exposure of F2 animals again to pathogen contributes another new 100 units of maternal product to F3 animals plus 1/100, after dilution, of the 101 total units it “built up” from P0 and F1 animals. Cumulatively, this would mean that if two generations of consecutive exposure of animals to pathogen contributes 101% of any hypothetical units to F3 offspring, then three consecutive generations contributes at maximum 101.01% of any hypothetical “built up” units to F3 offspring unless the units are either (1) not randomly diluted or (2) maintained in some way (Both possibilities would likely require an active mechanism). Finally, to affect F4 progeny survival as we report here, these units would have to be diluted 100-fold again between unexposed F3 animals and their F4 offspring such that two consecutive generations of exposure to *P. vranovensis* results in F3 animals contributing 1.01 units to F4 offspring, while three consecutive generations of exposure to *P. vranovensis* would result in F3 animals contributing 1.0101 units to F4 offspring.

It is difficult to imagine that an additional 0.01% contribution of some built-up P0 product explains the 3-fold+ enhancement of survival we observe in F4 progeny when three consecutive generations of animals are exposed to *P. vranovensis* when compared to only two consecutive generations animals being exposed to *P. vranovensis*. Therefore, we feel confident that our observed effect – even if the units mediating the observed effect are transferred maternally between each generation – represents a process that is distinct from a simple maternal build-up of a randomly diluted factor that takes two generations to dilute away. (Dilution likely still explains the eventual disappearance of the adaptive effect after 2 generations, but there is likely to be some mechanism that at least in part maintains the adaptive effect and lessens the effects of dilution each generation similar to small RNA inheritance where the original signal is amplified from one generation to the next but eventually the silencing dilutes away after several generations). We have added a discussion of this to the manuscript – lines 456-481.

Nonetheless, in an effort to avoid distracting or confusing any readers of the manuscript, we have changed the term transgenerational to multigenerational in most places in the text.

(Multigenerational being a more all-encompassing term that includes both intergenerational and transgenerational effects and is unencumbered by current debates over what qualifies as a “real” transgenerational effect).

Despite this change, we would like to note that a maternal effect that persists for multiple generations, similar to the one we report here, would be categorized as a transgenerational effect based on the definitions presented in several recent reviews of the field (example - Miska and Ferguson-Smith. (2016) *Science* 354(6308):59-63). More specifically, our observed effect, irrespective of the underlying mechanism, would be cumulatively defined as transgenerational because transgenerational effects were defined by phenotypic changes in F3 or later generation animals that are caused by changes in the environment of P0 animals, and not by the mechanism(s) by which such a phenotype might manifest. In addition, the term transgenerational was in part created to define effects that had no known mechanisms by which they might occur but simply could not be caused by (1) direct exposure of gametes that became F3 or later animals to the environment of P0 animals or (2) persistence of some randomly diluting factor. We are confident that the effect we report in Figure 2, while different from many observations of transgenerational effects that have been reported to date, meets this definition.

In summary, we are sympathetic to the fact that many different observations of transgenerational effects have been reported in diverse organisms that not uncommonly use different definitions of this term. We are confident that the results reported in Figure 2 fit within the umbrella of transgenerational, as opposed to intergenerational, for all of the reasons described above, but we have changed the term transgenerational to multigenerational in the abstract and text to avoid any possible distraction or confusion related to these terms. We do not believe such a change in terms in any way changes the novelty of or our excitement about the robust heritable adaptation to a natural pathogen we report here.

Again, a time-course with embryos laid at different timepoint might help in the interpretation of the results. Also, a better graphical representation of the experiment might be good to have it here (it is a bit confusing for how long the adults in each generation have been exposed to the pathogen - continuously or only for 24h in YA stage).

1.6 - We agree with Reviewer 1 and have added the graphical representation of the multigenerational experiment to the manuscript as Fig. 2c. In addition, we have added new data using staged embryos (See response 1.1). We believe these additions improve the clarity of our manuscript and further support our interpretations.

D. In figure 5C the authors show a transcriptional reporter of CYSL-2. This is a nice tool although a CRISPR-Cas9 tagged version of CYSL-2 (or RHY-1) can be used to monitor the persistence of the protein in the embryo and its persistence even in F1 adults after the 3 generations exposure. This will allow to monitor whether the maternal induced proteins can persist until the next generation. Also, it would be interesting to perform RNA FISH experiments to detect rhy-1, cysl-1 and cysl-2 and follow the accumulation of these RNAs in adult tissues and embryos.

1.7 – We agree that a CRISPR-Cas9 tagged version of CYSL-2 could be used to monitor the expression of and persistence of CYSL-2 in the embryo and have attempted to build such a strain, although we have so far not been able to create a version that is easily visible. We

have, however, added significant new data to the manuscript further profiling the expression of CYSL-1 and CYSL-2 using transcriptional reporters, including profiling the persistence of CYSL-2 in F1 animals (see response 2.2 for a full list of all the additional data added related to *cysl-1*, *cysl-2*, and *rhy-1* expression). Among these data we demonstrate that *cysl-2::GFP* is lost in F1 animals within 24 hours of clearing the infection. We believe this new data, in combination with other new data such as that described in response 1.5, suggests that the pathogen response and adaptation are quickly turned off if the pathogen infection is cleared.

E. The changes in gene expression in embryos upon adult exposure to *P. vranovensis* are really large compared to adult exposure to *P. aeruginosa* and *P. luminescence*. Yet, mutation in only one gene is sufficient to abolish this effect and its transgenic expression is able to rescue the phenotype. Are all those detected gene expression changes due to developmental differences in mixed stages embryo collected (most of the changed genes are very low expressed)?

1.8 - We have added substantial new data to the manuscript suggesting that many of the observed changes in gene expression are unlikely to be caused by differences in embryo staging. First, we now demonstrate that picking embryos of the same stage did not affect the observed adaptation (New Supplemental Fig. 2b – See response 1.2). Second, we have added new data indicating that many of the genes that exhibit the largest fold-change in response to *P. vranovensis*, including *cysl-1*, *cysl-2*, and *rhy-1* are known stress response genes regulated by signalling via the transcription factor SKN-1 and we confirmed this finding by performing RNA-seq of mutants with constitutively activated SKN-1 (New Supplemental Table 5). In support of a role for SKN-1 in regulating animals' response to infection, we have added new data demonstrating that partial loss-of-function mutations in *skn-1* (null mutations are embryonic lethal) exhibit reduced adaptation to *P. vranovensis* (New Fig. 3e). These results strongly support the model that signalling via SKN-1 regulates gene expression in response to *P. vranovensis*.

In addition to these data, we added new imaging of multiple independent GFP reporters for genes that exhibit increased expression in response to *P. vranovensis* by RNA-seq, including CYSL-2. We imaged similarly staged three-fold stage embryos from parents fed either *E. coli* or *P. vranovensis* and found that the difference in expression of these genes from parents fed *P. vranovensis*, when compared to *E. coli*, is not due to differences in embryo staging (New Supplemental Figs. 3 and 4 and new lines 371-383 of the text).

Collectively, these results indicate that many of the observed changes in gene expression in embryos from parents exposed to *P. vranovensis* are not simply due to differences in embryo staging and are likely at least in part due to differences in activation of the transcription factor SKN-1. We note that while many of the genes that exhibit the largest fold change in gene expression in response to *P. vranovensis* are normally lowly expressed, they are robustly expressed in response to *P. vranovensis* and the difference in fold change is not simply due to a low baseline. This pattern of expression is common amongst SKN-1 regulated stress response genes.

To reinforce their data the authors should perform RNA-seq in *rhy-1* mutant embryos exposed to the pathogen (Fig. 6b) and check how many of the upregulated genes are specific to the *rhy-1* gene activity. Also, the RNA-seq in the rescued embryos (*rhy-1* (n5500); *rhy-1* +) can further confirm the specific gene expression changes.

1.9 - In new data we have added to the manuscript, we found the *rhy-1* is itself a stress response gene that exhibits upregulated gene expression when the SKN-1 transcription factor is activated, as are *cysl-1* and *cysl-2*. Furthermore, we found that a partial loss-of-function allele (*skn-1* null mutants are maternal effect embryonic lethal) exhibited reduced adaptation to *P. vranovensis*. These results indicate that SKN-1 likely regulates the expression of a substantial fraction of the genes activated in response to *P. vranovensis*.

Based on these results, and in the spirit of Reviewer 1's suggestion, we sought to investigate how many of the genes that exhibit increased expression in response to *P. vranovensis* are dependent on the WDR-23/SKN-1 signalling pathway by RNA-seq. (We note that we do not believe *rhy-1*, which is itself an integral membrane protein of the endoplasmic reticulum, regulates the transcriptional response to *P. vranovensis*, but rather that it plays a role in the breakdown of toxins in line with as-yet-unpublished findings of Horsman et al., 2019 (*bioRxiv*) who also observed a *hif-1* independent function of *rhy-1* in mediating animal protection from hydrogen sulfide).

To investigate how many of the upregulated genes in response to *P. vranovensis* are regulated by the WDR-23/SKN-1 signalling pathway, we decided to perform RNA-seq of *wdr-23* mutant embryos which exhibit constitutively activated SKN-1. This is because *skn-1* null mutants are not viable and the weak loss-of-function allele of *skn-1* (a splice site allele) used in the new Fig. 3e is still capable of producing a fully functional copy of SKN-1 and partially adapting to *P. vranovensis*. Thus, this partial loss-of-function allele might not accurately reveal the fraction of genes that require proper SKN-1 activation for their expression. By performing RNA-seq of mutants with constitutively activated SKN-1, we found that a large fraction of genes that increased expression in response to *P. vranovensis* are also regulated by SKN-1 activity, including *rhy-1*, *cysl-1*, and *cysl-2*. We propose that many of the genes that exhibit increased expression in response to *P. vranovensis* are regulated by the WDR-23/SKN-1 signalling pathway in response to infection. This new data has been added to the text in new lines 347-433, which are substantially revised.

F. Is the colonization of adult worms by *P. vranovensis* required for the adaptive response to *P. vranovensis* in embryos? Maybe the author should check the colonization level of adult worms exposed to fluorescent *P. vranovensis* (using bacterial GFP reporter for example). In addition, because RHY-1 is a hypoxia inducible factor, the authors can try to expose the adult worms to hypoxia condition to see whether this is sufficient to trigger the adaptive responses to *P. vranovensis* in the embryos.

1.10 - We have added new data to the manuscript demonstrating that adding antibiotics to plates of infected *C. elegans* eliminates the adaptation observed in offspring. These results support the model that adult animals need to be fed live *P. vranovensis* to induce the adaptation in offspring.

With regards to hypoxia, our data demonstrate that mutants in the hypoxia signalling pathway (*egl-9* and *hif-1*) do not affect the adaptation to *P. vranovensis*. In addition, our data and others (such as Horsman et al., 2019 - <https://doi.org/10.1101/628784>) have indicated that RHY-1 is a more general stress response gene that is regulated by multiple stress pathways, such as SKN-1 signalling. For these reasons we have focused our manuscript on *C. elegans*' response to *P. vranovensis* and related pathogens to compare and contrast the specificity our response when compared to other pathogens.

Reviewer #2 (Remarks to the Author):

Burton et al. investigate the transgenerational heritability of pathogen defense response mechanisms in *C. elegans*. Recently, a number of publications have reported that *C. elegans* responds to the presence of pathogens by epigenetically regulating germline gene expression and that those epigenetic changes are inherited up to three consecutive generations. The topic of heritability of adaptive responses is highly interesting.

Burton et al. exposed worms to *P. vranovensis* and observe that the sensitivity of the animals is alleviated when the parental generation had been exposed to the pathogen. The authors should more clearly explain how those experiments were performed. They state in the methods part that embryos were collected. Here it is important to clarify whether bacterial contamination was excluded. Were the embryos collected by bleaching or just transferred by picking?

2.1- We agree with reviewer 2 and believe that adding a graphical representation of the experimental set up will substantially enhance the clarity of the manuscript and how the experiments were performed, in line with the suggestion of reviewer 1. These graphical representations have been added to the manuscript as Fig. 1a and Fig. 2e. In addition, we have added a graphical model summarizing the data reported in this manuscript as new Fig. 6.

We have also updated the methods to more clearly state that the embryos were collected by bleaching to eliminate bacterial contamination. Furthermore, we have added new data to the manuscript demonstrating that embryos of the same stage by picking results in the same observed adaptive effect as collecting embryos by bleaching, indicating that the method of embryo collection did not affect the observed adaptive effect (See response 1.2).

The authors report that culturing for multiple generations offers most protection and that a significant protection could be maintained to some degree when the last generation was exposed to non-pathogens. They then exclude known epigenetic regulators and also pmk-1 immune signaling as mediators of the transgenerational protection and instead perform gene expression analysis to identify genes responding to *P. vranovensis*. Here, they show that *cysl-1* and *cysl-2* as well as *rhy-1* are induced upon infection and required for the protection of offspring. The expression of those genes should be more thoroughly characterized. Where and when are they expressed? Only in embryos or in the parents? In which tissues/cell types? When are they switched off, are the pmk-1 dependent?

2.2 - We have added several pieces of new data investigating the expression of genes in response to *P. vranovensis* infection that substantially enhance our understanding of *C. elegans* response to *P. vranovensis*. We have summarized these additions below:

- (1) Further analysis comparing the transcriptional responses of adults to *P. vranovensis* in comparison to embryos from parents exposed to *P. vranovensis*. This analysis was based on RNA-seq data. We found that many, but not all, of the genes that exhibit increased expression in response to *P. vranovensis* are upregulated in both adults and in embryos from infected adults. We have added this data to the text as new lines 294-305. In addition, we have data to the supplementary tables for this manuscript listing (1) genes upregulated only in adult animals (2) genes upregulated in embryos from

infected animals but not in adults, and (3) genes upregulated in both adults and embryos from infected adults. We note that *cysl-1*, *cysl-2*, and *rhy-1* are upregulated in both adults and in embryos.

- (2) We have added a more in-depth investigation of *cysl-1* and *cysl-2* expression when feeding on either *E. coli* or *P. vranovensis* using GFP reporters. Briefly, we found that *cysl-1* is predominantly expressed in hypodermal cells and neurons in larvae, but that hypodermal expression is largely lost in adult animals. This expression was not significantly affected by *P. vranovensis* exposure, possible because the multicopy transgenic reporter is already highly overexpressed even in animals feeding on *E. coli*. By contrast, we found that *cysl-2* is only lowly expressed in larvae and adults feeding on *E. coli* (except in the pharynx where GFP expression appears bright), but exhibits a dramatic increase in expression in response to *P. vranovensis*. Expression of *cysl-2* in response to *P. vranovensis* is predominantly observed in hypodermal tissue. These data have been added to the manuscript as new supplemental figures 3 and 4 and to the text as new lines 399-417.
- (3) We show that the expression of *cysl-1*, *cysl-2*, and *rhy-1* are all regulated by signalling via the WDR-23/SKN-1 stress response pathway and that partial loss-of-function *skn-1* mutants (null mutants are not viable) exhibit reduced adaptation to *P. vranovensis*. See responses 1.8 and 1.9.

We believe these data substantially enhance our knowledge of when and where *cysl-1*, *cysl-2*, and *rhy-1* are expressed in response to *P. vranovensis* and the signalling pathways that control their expression.

The experiments appear to be well performed and the authors' conclusions are backed by the data. They identify with *cysl-1/-2* and *rhy-1* a potentially new mechanism of transgenerational pathogen resistance that appears to be distinct from the recently established epigenetic mechanisms. It remains unclear how those cysteine synthases and regulator of hypoxia inducible factor RHY-1 operate in this context. No mechanistic insight into their biochemical or regulatory function and the phenotypic consequence is probed for. How is cysteine metabolism related to this? Is there any role for hypoxia? Is this a general stress response or a specific pathogen response?

2.3 – We thank Reviewer 2 for the kind comments regarding the experiments and conclusions reported in this manuscript. We have now added additional discussion of how CYSL-1, CYSL-2, and RHY-1 might operate mechanistically to mediate this adaptation to the text as part of a substantially revised discussion section (see new lines 436-537) and added a new graphical model summarizing the observations and mechanistic data reported in this manuscript (new Fig. 6).

Briefly, we note that cysteine synthases, such as CYSL-1 and CYSL-2 have previously been reported to be involved in breaking down bacterial toxins, in addition to their role in cysteine metabolism (Budde and Roth, *Genetics* 2011). Similarly, in as-yet-unpublished results recently posted to *bioRxiv*, RHY-1 was demonstrated to function independently of its known role in hypoxia to mediate resistance to toxic amounts of hydrogen sulfide (Horsman et al., 2019 - <https://doi.org/10.1101/628784>).

We propose that CYSL-1, CYSL-2, and RHY-1 are likely required for resistance to as-yet-unknown bacterially produced toxins (pathogenic species of *Pseudomonas* are known to produce an incredibly diverse array of potentially toxic chemicals) and text has been updated to discuss possible mechanism (new lines 436-454). Furthermore, our data suggest that this effect is likely specific to only certain pathogens or toxins as we found that exposure to *P. aeruginosa* and *P. luminescens* did not result in the same adaptation.

Reviewer #3 (Remarks to the Author):

Burton et al - *C. elegans* heritably adapts to *P. vranovensis* infection via a mechanism that requires the cysteine synthases CYSL-1 and CYSL-2 ‘

Here the authors examine the effects of the soil bacteria *P. vranovensis*, which they identified as a natural bacterial isolate, on *C. elegans* and its offspring. They find that a single generation of exposure to Pv did not induce the multigenerational effect, but rather only acts intergenerationally, unless treated consecutively from P0 through F3, the F5 generation survived longer. They find that many genes involved in transgenerational inheritance from a short exposure to bacteria are not involved; instead, PMK-1 is required for survival after Pv exposure, but not for the adaptation effect. Other pathogens did not induce adaptation to Pv (see note below). Gene expression and metabolic profiling were carried out. The cysteine synthases CYSL-1 and -2 are increased in embryos exposed to Pv, and mutants in *cysl-1* and *cysl-2* are unable to adapt to Pv; similarly *rhy-1* mutants did not adapt, but *hf-1* and *egl-9* mutants do, suggesting the pathways are separate.

The authors discuss differences with *Pseudomonas aeruginosa* avoidance, which is transgenerational, vs offspring survival on Pv after 4 generations of exposure, which is more similar to dauer induction after P0-F2 exposure to PAO1 and *S. enterica*.

This paper is somewhat inconclusive, in that it points out some differences with other paradigms, but doesn't clearly identify a distinguishing pathway or signaling mechanisms. It does have some strengths, however (interesting effect, clearly different from transgenerational Pa avoidance, similarities to dauer induction after multigenerational exposure) so it would be interesting to see what the authors might be able to further flesh out mechanistically.

We thank reviewer 3 for their kind comments regarding our interesting effect and highlighting the clear differences from previous observations. We have added new mechanistic data to the manuscript indicating that our observed heritable adaptation is likely regulated by signalling via the stress-response transcription factor SKN-1. We believe these data significantly enhance our manuscript and establish some mechanistic data for why adaptations to *P. vranovensis* might be different from those reported for *P. aeruginosa*.

Major comments:

1. It seems like an exaggeration to write “These results indicate that the environment of P0 animals can enhance the survival of F4 generation” since it is really more correct to write P0 through F2 (three generations)- it is not the P0 environment, but rather that plus the next two generations of exposure that induces the observed increase in offspring survival.

3.1 - We believe our results do demonstrate that the exposure of P0 animals to *P. vranovensis* is required, but not sufficient on its own, for the enhanced survival observed in F4 animals in the experimental set up we tested, but our text likely caused confusion on this topic. We have

modified the text in question to more clearly describe our observed results as -- “These results indicate that the exposure of P0 animals to *P. vranovensis* is required to enhance the survival of F4 generation animals under conditions where F1 and F2 animals are also exposed to *P. vranovensis* (Fig. 2c)” (New lines 247-250).

For reference, our results in Fig. 2b demonstrate that exposing P0 animals to *E. coli* and F1 and F2 animals to *P. vranovensis* results in F4 animals that exhibited under 1% survival in response to *P. vranovensis*. By contrast, we found that exposing P0, as well as F1 and F2, animals to *P. vranovensis* results in approximately 10% of F4 animals surviving exposure to *P. vranovensis*. Because the F1-F3 generation’s environments are held constant in these experiments, our observations indicate that the P0 exposure to pathogen is required, but not sufficient on its own, to increase F4 survival to approximately 10%. We believe this is an effect of the P0 environment that enhances survival of the F4 generation, even if P0 exposure alone is not sufficient to affect the F4. To better communicate our observations, we have also added a new graphical representation of these results as new Fig. 2c, rewritten the text on this section as new lines 240-255, and added a more in-depth discussion of the results as lines 456-481.

2. The logic of exposing the worms to two other pathogens, then testing offspring survival to Pv seems turned around. First, the authors should test whether the other pathogens induce a similar effect as Pv does (i.e., does P0-F2 exposure to PA14 increase offspring survival in the F4 generation exposed to PA14?), and whether multigenerational Pv exposure induces offspring survival to the other pathogens – these will better answer the question posed: whether heritable adaptation to Pv is a general or specific response.

3.2 - We thank the reviewer for pointing out this lack of clarity on why we ran the experiments we did, and we have significantly rearranged this portion of the text to better clarify why we tested the specific species of bacteria we did and the intention of these specific experiments. Briefly, our motivation for comparing to *P. aeruginosa* and *P. luminescens* was based on two observations.

First, we performed a comparison of the transcriptional response to *P. vranovensis* to all other *C. elegans* transcriptional responses to microbes that have previously been studied using WormExp v1.0. (A description of this has now been added to the text – new lines 308-317). We found that the transcriptional response to *P. vranovensis* is most similar to the response to *P. aeruginosa* and *P. luminescens*.

Second, a very recent study (Moore et al., 2019 Cell, 177, 1827-1841) found that parental exposure of *C. elegans* to *P. aeruginosa* enhances offspring survival in response to future exposure to *P. aeruginosa* via a mechanism that requires the argonaute PRG-1. This study already determined that parental exposure to PA14 can affect offspring survival. Furthermore, we have also found that parental exposure of *C. elegans* to *P. aeruginosa* affects the developmental rate of offspring in response to future exposure to *P. aeruginosa* (Burton et al., 2017). These studies collectively demonstrate that parental exposure of *C. elegans* to *P. aeruginosa* can affect progeny survival, behaviour, and development in response to future exposure to *P. aeruginosa*, although we note that the effects of P0 exposure to *P. aeruginosa* on progeny survival and development (Moore et al., 2019) are very modest when compared to the survival differences we see with *P. vranovensis*.

Based on these two observations, we performed the specific assays we did to (1) determine if our observed heritable adaptation to *P. vranovensis* was mechanistically the same as or different from previous studies of *P. aeruginosa* and (2) to determine if exposure of parents to other pathogens that promote a similar transcriptional changes were sufficient to promote adaptation in offspring or if our observed adaptive effect is specific to *P. vranovensis*.

We found (1) parental exposure to *P. aeruginosa* and *P. luminescens* did not protect offspring from *P. vranovensis*, (2) parental exposure to *P. aeruginosa* and *P. luminescens* did not result in the same changes in gene expression as parental exposure to *P. vranovensis*, and (3) PRG-1 is not required for adaptation to *P. vranovensis*.

We believe these experiments are sufficient to demonstrate that our findings are (1) different from those previously reported for *P. aeruginosa* in both magnitude and in mechanism and (2) that our observed adaptation to *P. vranovensis* is specific to *P. vranovensis* and not general since *P. aeruginosa* and *P. luminescens* did not induce this adaptation. We believe these observations answer the questions we set out the answer. We also did not further pursue studies of *P. aeruginosa* or *P. luminescens* in this manuscript as our data indicated that the response of *C. elegans* to these other pathogens is distinct from its response to *P. vranovensis*.

3. Was anything learned from the gene expression profiling that is described before the metabolomics analysis? This seems like a lost opportunity.

3.3 - We agree that more could be learned from investigating the gene expression profiling and we have added substantial new data related to gene expression profiling to the manuscript – See response 2.2.

We believe these data significantly enhance our understanding of *C. elegans* response to this pathogen.

4. The study does not identify the reason (underlying mechanism) for the differences in adaptation to Pv vs Pseudomonas aeruginosa.

3.4 - Here we describe that *C. elegans*' adaptation to *P. vranovensis* requires the cysteine synthases CYSL-1 and CYSL-2 and the stress response gene RHY-1. Furthermore, we have added new mechanistic data to this revised manuscript demonstrating that the stress response transcription factor SKN-1 is required for full adaptation to *P. vranovensis* and likely regulates a substantial amount of the transcriptional response to *P. vranovensis* (New Fig. 3 and new lines 348-434). By contrast, Moore *et al.*, reported that the adaptation to *P. aeruginosa* requires the argonaute PRG-1. These findings are themselves mechanistic in nature and the differences in factors and signalling pathways required might explain why these two adaptations appear to be phenotypically distinct.

We agree with Reviewer 3 that future studies of these two paradigms to identify the complete mechanisms by which these adaptations occur will be of high interest; neither adaptation is completely mechanistically solved at this point. Nonetheless, we believe that all of the claims made in this manuscript are well supported, provide initial mechanistic insight into the differences between these two adaptations, and will be of significant interest to researchers studying host-pathogen interactions, researchers studying intergenerational and transgenerational effects, and the readers of *Nature Communications*.

5. In the Discussion, the authors compare the differences in responses to Pv and Pa, but the assays used for the two are different (avoidance to Pa vs offspring survival on Pv), so it is difficult to even make these comparisons.

3.5 - We have updated the discussion to note that offspring survival was also assayed for *P. aeruginosa* by Moore et al., (2019) and that they similarly observed that parental exposure to *P. aeruginosa* enhances offspring survival in response to future *P. aeruginosa* exposure (new lines 486-488), albeit they observe a much more modest effect on survival than we observe for *P. vranovensis* (See response 3.2). Notably, the assays of survival in response to these two species of *Pseudomonas* are the same and both studies investigate proposed transgenerational effect of infection. Thus, we believe it is reasonable to compare and contrast the differences in adaptation to *P. aeruginosa* and *P. vranovensis* as part of a discussion on where our data fits in the broader field.

Furthermore, we have added data to the manuscript demonstrating that when we compare the transcriptional response to *C. elegans* to *P. vranovensis* to all previous gene expression profiles of *C. elegans* responses to microbes using WormExp v1.0, we found that *C. elegans* response to *P. vranovensis* is most similar to *C. elegans* response to *P. aeruginosa*. This observation further supports the usefulness in determining if *C. elegans*' responses to these two *Pseudomonas* pathogens are the same or different.

With that said, we'd be happy to shorten the discussion of these two studies to simply state that the two adaptations appear to be mechanistically distinct should the reviewers prefer a shorter comparison.

6. No clear mechanistic model emerges, just the involvement of two cysteine synthases and RHY-1, but these roles are not well described either, beyond being required.

3.6 - We have expanded upon our discussion section to include a proposal of what we believe to be the likely role of the cysteine synthases and RHY-1 in mediating adaptation to *P. vranovensis* -- new lines 437-455. We have also added a new graphical model summarizing the adaptive response and the mechanistic data reported in this manuscript -- new Fig. 6. See response 2.2 for a more detailed description of the proposed mechanism by which these genes act.

Briefly, previous studies found that cysteine synthases can break down toxins produced by certain pathogens, in addition to their role in cysteine metabolism (Budde and Roth, *Genetics* 2011). Similarly, as-yet-unpublished studies recently uploaded to bioRxiv (Horsman et al., 2019 - <https://doi.org/10.1101/628784>) found that RHY-1 and CYSL-1 mediate resistance to hydrogen sulfide. We believe that the cysteine synthases and RHY-1 likely play a similar role in the response to *P. vranovensis* and are required to break down as-yet-unknown bacterially produced toxins.

Minor comments:

1. The authors jump from describing BIGb446 and 468 to discussing *P. vranovensis* before the strains were identified as Pv (p. 5, line 137).

3.7 - We thank the reviewer for this comment and have updated this line of text accordingly.

REVIEWERS' COMMENTS:

Reviewer #1 (Remarks to the Author):

In this revised manuscript Burton et al. have addressed almost all major points and they greatly improved the clarity of the manuscript. However, there is still one point that is not clear to me. I originally thought to the possibility that the maternal effect would be caused by the inheritance (through the germline) of stressed-induced proteins (or RNA) such as CYSL-1, CYSL-2 and their persistence in the embryo. Indeed, they have recently shown how the inheritance of pathogen avoidance behavior depends on germline mechanisms (piRNAs). This is why I was asking the authors to build a CRISPR-Cas9 reporter of these proteins and/or perform RNA smFISH. Unfortunately, the multicopy transgenic array (which is most likely silenced in the germline) cannot be used to test this hypothesis.

Nonetheless, the data shown in Supplementary Fig. 5, 6 are also suggesting an additional hypothesis. In supplementary Fig. 5b, CYSL-2 transcriptional reporter is induced by *P. vranovensis* exposure (in Adult). However, in the embryo the transcriptional reporter started to be activated again after gastrulation only in the progeny previously exposed to *P. vranovensis* and not in the control (Supplementary Fig. 5d). Therefore, it is unlikely that the activation of the transgene in the embryo is caused by the exposure to the pathogen on the plate. Thus, this result poses an additional question: how these genes are transcriptionally activated again in the embryo? Are chromatin mechanisms involved? The authors propose in their model that the transcriptional factor SKN-1 is the mediator of the heritable adaptation to the pathogen by activating CYSL-1, CYSL-2, RHY-1 and other stress responsive genes. It is known, that SKN-1 is maternally inherited in the embryo and is essential during embryogenesis. Thus, it is unlikely that the differences in the transcriptional activation in the embryo are due to differential loading of maternal SKN-1. Therefore, it might be possible that heritable chromatin changes are responsible for the reactivation of those genes in the embryo. This is something that in my opinion the authors should consider and somehow address or comment. For instance, I have noticed that in Supplementary Fig. 3a some chromatin factor mutants might have some statistically significant effects, similar to the effect shown for SKN-1 loss of function (Supplementary Fig. 3e). I wonder if they can conduct some statistical tests to see whether some of these chromatin factors show some significant effects and whether they can also perform a similar assay they have done for *skn-1* in Fig. 3f (note: the “f” is not present in the figure). Of course, the effect after 72 hrs might also be uncoupled by “heritable” effects and being just caused by the lack of stress induced gene expression program.

Finally, I think it would be nice to add some comments on the possible germline mechanisms involved in the transmission of the pathogen resistance effect described in this manuscript. In their discussion they have excluded mechanisms such as piRNAs and chromatin factors, however they haven't proposed any alternative mechanisms to explain this heritable effect (except that their effect is due to the increased expression of pathogen-induced genes), and they cannot exclude that chromatin changes might be involved.

Reviewer #2 (Remarks to the Author):

The authors have now better explained their experimental procedures and further confirmed the intergenerational effect. They also included data on the expression of *cysl-1* and *cysl-2* in vivo. Also the discussion is now more comprehensive. In conclusion, the authors present a new type of intergenerational pathogen resistance that should be of broad interest. I am fully satisfied with the revisions.

Björn Schumacher

Reviewer #3 (Remarks to the Author):

This is a much improved version of the manuscript that has addressed most of my previous concerns.

1. Thank you for changing “transgenerational” to “multigenerational” and for elucidating the distinctions between recent studies of PA14 avoidance and this Pv adaptation. Likely both mechanisms are at work but are quite distinct, as the authors note.
2. Figure 1a would be more informative if the two bacteria were different colors (they look the same on this computer). Does the figure also accurately represent the spreading on the plate (small spot vs whole plate)? Same for Figure 2c.
3. Since lethality of bacteria varies significantly with temperature, please state in the main text exactly what temperature the assays are done at, since lethality of bacteria varies significantly with temperature. What is room temperature in this lab? 20°C? 18°C? 22°C? Without this information, it is hard to make comparisons with other published work.
4. Survival: please report Kaplan-Meier statistics for survival (not just a single time point).
5. I think the dilution discussion should be deleted or made less specific. Without knowing exactly what the mechanism in the germline is that allows this multigenerational effect, the calculation could be way off. Once the factor(s) is/are known, and can be measured, then such a calculation could be made.

Minor:

- missing period at end of last sentence in introduction

Reviewer #1 (Remarks to the Author):

In this revised manuscript Burton et al. have addressed almost all major points and they greatly improved the clarity of the manuscript.

We thank reviewer 1 for their kind comments.

However, there is still one point that is not clear to me. I originally thought to the possibility that the maternal effect would be caused by the inheritance (through the germline) of stressed-induced proteins (or RNA) such as CYSL-1, CYSL-2 and their persistence in the embryo. Indeed, they have recently shown how the inheritance of pathogen avoidance behavior depends on germline mechanisms (piRNAs). This is why I was asking the authors to build a CRISPR-Cas9 reporter of these proteins and/or perform RNA smFISH. Unfortunately, the multicopy transgenic array (which is most likely silenced in the germline) cannot be used to test this hypothesis.

Nonetheless, the data shown in Supplementary Fig. 5, 6 are also suggesting an additional hypothesis. In supplementary Fig. 5b, CYSL-2 transcriptional reporter is induced by *P. vranovensis* exposure (in Adult). However, in the embryo the transcriptional reporter started to be activated again after gastrulation only in the progeny previously exposed to *P. vranovensis* and not in the control (Supplementary Fig. 5d). Therefore, it is unlikely that the activation of the transgene in the embryo is caused by the exposure to the pathogen on the plate. Thus, this result poses an additional question: how these genes are transcriptionally activated again in the embryo? Are chromatin mechanisms involved? The authors propose in their model that the transcriptional factor SKN-1 is the mediator of the heritable adaptation to the pathogen by activating CYSL-1, CYSL-2, RHY-1 and other stress responsive genes. It is known, that SKN-1 is maternally inherited in the embryo and is essential during embryogenesis. Thus, it is unlikely that the differences in the transcriptional activation in the embryo are due to differential loading of maternal SKN-1. Therefore, it might be possible that heritable chromatin changes are responsible for the reactivation of those genes in the embryo. This is something that in my opinion the authors should consider and somehow address or comment. For instance, I have noticed that in Supplementary Fig. 3a some chromatin factor mutants might have some statistically significant effects, similar to the effect shown for SKN-1 loss of function (Supplementary Fig. 3e). I wonder if they can conduct some statistical tests to see whether some of these chromatin factors show some significant effects and whether they can also perform a similar assay they have done for *skn-1* in Fig. 3f (note: the “f” is not present in the figure). Of course, the effect after 72 hrs might also be uncoupled by “heritable” effects and being just caused by the lack of stress induced gene expression program. Finally, I think it would be nice to add some comments on the possible germline mechanisms involved in the transmission of the pathogen resistance effect described in this manuscript. In their discussion they have excluded mechanisms such as piRNAs and chromatin factors, however they haven’t proposed any alternative mechanisms to explain this heritable effect (except that their effect is due to the increased expression of pathogen-induced genes), and they cannot exclude that chromatin changes might be involved.

We agree with reviewer 1 that it is very interesting that the expression of genes such as *cysl-2* occurs only after gastrulation in the embryos from parents exposed to *P. vranovensis*. In addition, we agree that the precise mechanism by which this reactivation of gene expression occurs is unclear. We have therefore added three possible

explanations for this result to the discussion, which includes the possibility of chromatin related factors. We note, however, that none of the chromatin factors we tested exhibit statistically significant effects (*spr-5* is close at $p = 0.07$ but did not meet the cut off for significant). Furthermore, we note that some of the mutants in chromatin factors are themselves “sick” and it’s possible that some of the reduced survival might be due to indirect effects caused by the disruption of other processes.

Briefly, the possibilities added to the manuscript are (lines 632-643):

- (1) It is possible that changes in chromatin marks at the *cysl-2* loci explain the change in gene expression in offspring.
- (2) A second possibility is that there are multiple *skn-1* isoforms (SKN-1a/b/c) produced by alternative start codons and that differential loading of specific *skn-1* isoforms or post-translationally modified versions of the SKN-1 protein explain the change in expression. (SKN-1 can be phosphorylated at multiple sites).
- (3) A third possibility is that a specific signal, hormone, metabolite, or toxin is differentially deposited into embryos that results in a change in gene expression that occurs during development.

We believe the addition of this discussion will shed light on some of the possible explanations for our newly described heritable effect on gene expression and believe that future studies that can resolve which of these possibilities might explain the observed heritable change in gene expression will be of high interest.

Reviewer #2 (Remarks to the Author):

The authors have now better explained their experimental procedures and further confirmed the intergenerational effect. They also included data on the expression of *cysl-1* and *cysl-2* in vivo. Also the discussion is now more comprehensive. In conclusion, the authors present a new type of intergenerational pathogen resistance that should be of broad interest. I am fully satisfied with the revisions.
Björn Schumacher

We thank reviewer 2, Dr. Schumacher, for their kind comments.

Reviewer #3 (Remarks to the Author):

This is a much improved version of the manuscript that has addressed most of my previous concerns.

1. Thank you for changing “transgenerational” to “multigenerational” and for elucidating the distinctions between recent studies of PA14 avoidance and this Pv adaptation. Likely both mechanisms are at work but are quite distinct, as the authors note.

We are glad this change is appreciated.

2. Figure 1a would be more informative if the two bacteria were different colors (they look the same on this computer). Does the figure also accurately represent the spreading on the plate (small spot vs whole plate)? Same for Figure 2c.

We have changed the colors to make the difference between the two bacteria greater, however we have not made the colors blue/yellow or a different pairing as we wanted to retain a rough approximation of the actual colors of the bacteria as opposed to other bacterial pathogens, such as *P. aeruginosa*, which can be blue.

3. Since lethality of bacteria varies significantly with temperature, please state in the main text exactly what temperature the assays are done at, since lethality of bacteria varies significantly with temperature. What is room temperature in this lab? 20°C? 18°C? 22°C? Without this information, it is hard to make comparisons with other published work.

We have revised the text to state that room temperature in this lab is 22°C.

4. Survival: please report Kaplan-Meier statistics for survival (not just a single time point).

The data presented in Fig. 1c for a survival curve is the averages at each time point of a population of animals from three separate experiments and thus as a collection of percentages is not analysable by Kaplan-Meier which measures individual events. However, we have added the p-value of the individual experiments calculated by Kaplan-Meier which was $p < 0.001$ using Prism software analysis to do log rank tests. This statistical information and value were added to the figure legend.

5. I think the dilution discussion should be deleted or made less specific. Without knowing exactly what the mechanism in the germline is that allows this multigenerational effect, the calculation could be way off. Once the factor(s) is/are known, and can be measured, then such a calculation could be made.

We are happy to remove this section and have deleted it from the final text.

Minor:

- missing period at end of last sentence in introduction

We have confirmed a period in the last sentence of the introduction.